# Oncolytic H-1 Parvovirus Hijacks Galectin-1 to Enter Cancer Cells

**DOI:** 10.3390/v14051018

**Published:** 2022-05-11

**Authors:** Tiago Ferreira, Amit Kulkarni, Clemens Bretscher, Petr V. Nazarov, Jubayer A. Hossain, Lars A. R. Ystaas, Hrvoje Miletic, Ralph Röth, Beate Niesler, Antonio Marchini

**Affiliations:** 1Laboratory of Oncolytic Virus Immuno-Therapeutics, German Cancer Research Centre, Im Neuenheimer Feld 242, 69120 Heidelberg, Germany; bt_tiago@hotmail.com (T.F.); c.bretscher@gmx.de (C.B.); 2Laboratory of Oncolytic Virus Immuno-Therapeutics, Luxembourg Institute of Health, 84 Val Fleuri, L-1526 Luxembourg, Luxembourg; amitkulkar@gmail.com; 3Bioinformatics Platform and Multiomics Data Science Research Group, Department of Cancer Research, Luxembourg Institute of Health, L-1526 Luxembourg, Luxembourg; petr.nazarov@lih.lu; 4Department of Biomedicine, University of Bergen, 5007 Bergen, Norway; jubayer.hossain@uib.no (J.A.H.); larsystaas@gmail.com (L.A.R.Y.); hrvoje.miletic@uib.no (H.M.); 5Department of Pathology, Haukeland University Hospital, 5021 Bergen, Norway; 6nCounter Core Facility, Institute of Human Genetics, University of Heidelberg, 69120 Heidelberg, Germany; ralph.roeth@med.uni-heidelberg.de (R.R.); beate.niesler@med.uni-heidelberg.de (B.N.); 7Department of Human Molecular Genetics, University of Heidelberg, 69120 Heidelberg, Germany

**Keywords:** oncolytic virus immunotherapy, protoparvovirus H-1PV, virus host interactions, virus cell entry, galectin-1, laminin γ1

## Abstract

Clinical studies in glioblastoma and pancreatic carcinoma patients strongly support the further development of H-1 protoparvovirus (H-1PV)-based anticancer therapies. The identification of cellular factors involved in the H-1PV life cycle may provide the knowledge to improve H-1PV anticancer potential. Recently, we showed that sialylated laminins mediate H-1PV attachment at the cell membrane. In this study, we revealed that H-1PV also interacts at the cell surface with galectin-1 and uses this glycoprotein to enter cancer cells. Indeed, knockdown/out of *LGALS1,* the gene encoding galectin-1, strongly decreases the ability of H-1PV to infect and kill cancer cells. This ability is rescued by the re-introduction of *LGALS1* into cancer cells. Pre-treatment with lactose, which is able to bind to galectins and modulate their cellular functions, decreased H-1PV infectivity in a dose dependent manner. In silico analysis reveals that *LGALS1* is overexpressed in various tumours including glioblastoma and pancreatic carcinoma. We show by immunohistochemistry analysis of 122 glioblastoma biopsies that galectin-1 protein levels vary between tumours, with levels in recurrent glioblastoma higher than those in primary tumours or normal tissues. We also find a direct correlation between *LGALS1* transcript levels and H-1PV oncolytic activity in 53 cancer cell lines from different tumour origins. Strikingly, the addition of purified galectin-1 sensitises poorly susceptible GBM cell lines to H-1PV killing activity by rescuing cell entry. Together, these findings demonstrate that galectin-1 is a crucial determinant of the H-1PV life cycle.

## 1. Introduction

Oncolytic viruses selectively infect and destroy cancer cells while sparing normal tissues [1]. They can also stimulate strong anti-tumour immune responses and destroy tumour vasculature [2]. No fewer than 40 oncolytic viruses are currently under evaluation in clinical trials as treatments against a variety of cancers. Among them is H-1 rat protoparvovirus (H-1PV), a member of the *Parvoviridae* family in the genus *Protoparvovirus* [3,4]. This genus in addition to H-1PV includes Kilham rat virus, LuIII virus, minute virus of mice (MVM), mouse parvovirus, tumour virus X and rat minute virus [5,6], which are also evaluated at the preclinical level as oncolytic viruses.

The H-1PV genome is a linear, single-stranded DNA molecule of around 5 kb containing the P4 and P38 promoters. The P4 promoter regulates the expression of the non-structural (NS) gene unit encoding the NS1 and NS2 proteins. The P38 promoter controls the expression of the structural VP gene unit, which encodes the VP1 and VP2 capsid proteins and the non-structural small alternatively translated protein [4]. NS1 is the major regulator of viral DNA replication and gene transcription and is the major effector of virus oncotoxicity [7,8].

Preclinical studies in a number of cellular and animal models indicate that H-1PV can target a large variety of tumour cell lines from different tumour entities [4]. This preclinical evaluation paved the way for the clinical evaluation of H-1PV in patients with glioblastoma (GBM) [9] or pancreatic carcinoma [10]. In early-phase clinical trials, H-1PV treatment was shown to be safe and well-tolerated. Virus treatment was also associated with the first evidence of efficacy, including (i) the ability to cross the blood–brain barrier after intravenous delivery; (ii) effective distribution and expression in the tumour bed; (iii) immunoconversion of the tumour microenvironment; and (iv) improved progression-free survival and overall survival in comparison to historical controls [9]. However, treatment with H-1PV, like other oncolytic viruses, was unable to eradicate tumours in patients with the regimes used [9]. Therefore, there is an urgent need to improve the clinical outcome of H-1PV oncolytic therapy. A promising approach is the identification of host cell factors that modulate the H-1PV life cycle. This knowledge could provide hints to which drugs or treatment modalities might be combined with the virus in order to enhance its oncotoxicity. In addition, a deeper understanding of the H-1PV life cycle could help to identify biomarkers capable of predicting which patients would most likely benefit from virus treatment [11].

The first step of the virus life cycle is the virus recognition of receptor (s), co-receptor (s) or other co-factors on the cell surface modulating host cell entry. In the case of H-1PV and other protoparvoviruses, sialic acid is essential for virus–cell attachment [12,13,14]. Recently, we performed a druggable genome-wide siRNA library screen to identify putative modulators of H-1PV infection. The screen identified *LAMC1*, encoding the laminin γ1 chain, as a positive modulator of virus transduction. Characterisation of the interaction between H-1PV and laminin γ1 revealed that laminins, and in particular those containing laminin γ1, play a key role in mediating H-1PV attachment at the cell surface and subsequent entry into cancer cells. H-1PV binding to laminin is dependent on the sialic acid moieties in these molecules [14]. We have also shown that H-1PV cell uptake occurs through clathrin-mediated endocytosis and that the virus then passes through early to late endosomes prior to entering the nucleus. These events are dependent on dynamin activity and low endosomal pH [15].

The siRNA library screen also identified *LGALS1,* the gene encoding galectin-1 (Gal-1), as a leading activator of H-1PV infection. Interestingly, galectins are known to interact with laminins [16,17]. To date, 15 galectins have been identified in mammals [18]. They are widely expressed in various cell types and are involved in a variety of physiological functions including cell migration, mediation of cell–cell interactions, cell–matrix adhesion, transmembrane signalling, inflammation, and the immune response [19]. All galectins share a highly conserved carbohydrate-recognition domain, which binds to β-galactosides in N-linked and O-linked glycoproteins [20]. However, despite their similarities, galectins have notably different binding properties. Galectins are increasingly recognised as mediators of viral infections. However, the specific outcome of a galectin-virus interaction depends heavily on the particular galectin, the cell type, the virus, and the surrounding microenvironment. For instance, Gal-1 stabilises the binding of the human immunodeficiency virus (HIV)-1 to the host CD4 receptor on the surface of T cells by crosslinking CD4 and viral gp120 [21]. By contrast, Gal-1 inhibits Influenza A virus infection by interacting directly with the viral envelope glycoproteins [22]. In the context of protoparvoviruses, Gal-3 promotes MVM cell uptake and infection [23,24].

In view of the potential use of H-1PV as an anti-cancer therapeutic, our goal is to characterise the early events of H-1PV infection. In this study, we demonstrate that Gal-1 plays a key role in H-1PV infection at the level of virus entry.

## 2. Materials and Methods

### 2.1. Cells

Cervical carcinoma-derived HeLa, pancreatic ductal adenocarcinoma-derived BxPC3, glioma-derived, NCH125, NCH37, U251, LN308, T98G, and A172-MG cell lines were maintained in-house [14]. The NCH125 LGALS1 KO and NCH125 CRISPR Control cell lines were established in this study (see below-*Generation of LGALS1 knockout cell line*). HeLa Control and HeLa LAMC1 KD cells were established in a previous study [14]. All cells were cultured in Dulbecco’s modified Eagle’s medium (DMEM) supplemented with 10% FBS, 100 units/mL penicillin, 100 μg/mL streptomycin and 2 mM L-glutamine (all from Gibco, Thermo Fischer Scientific, Darmstadt, Germany) in a humidified incubator at 37 °C and 5% CO_2_. All cancer cell lines were regularly tested for mycoplasma contamination using a VenorGEM OneStep Mycoplasma contamination kit (Minerva Biolabs, Berlin, Germany) and tested by a human cell authentication test (Multiplexion GmbH, Mannheim, Germany).

### 2.2. Viruses

Both wild-type H-1PV and recombinant H-1PV harbouring the green fluorescent protein-encoding gene (recH-1PV-EGFP) were produced, purified and titrated as previously described [25,26].

### 2.3. siRNA-Mediated Knockdown

Cells were seeded at a density of 4 x 10^4^ cells/well in 24-well plates and grown in 500 µL of normal growth medium. After 24 h, cells were transfected with 10 nM siRNA using Lipofectamine RNAiMAX (Thermo Fisher Scientific, Carlsbad, CA, USA) according to the manufacturer’s instructions. The following siRNAs were used for the galectins study (all purchased from Life Technologies, Paisley, UK: Silencer Select *LGALS1* siRNA (Cat. N. 4390824), Silencer *LGALS3* siRNA (Cat. N. 11332) and Silencer Select Negative Control #2 siRNA (Cat. N. 4390846). The siRNA targeting the *LAMC1* gene (Cat. N. SI00035742) and the AllStars Negative siRNA (Cat. N. SI03650318) used as a negative control were purchased from Qiagen (Hilden, Germany). After 24 h, the medium was replaced, and cells were grown for an additional 24 h to allow efficient gene silencing.

### 2.4. Viral Transduction Assay

Depending on the experiment, after 48 h siRNA transfection or 24 h after seeding or after pre-treatment with chemical for 30 min, cells were infected for 24 h with recH-1PV-EGFP at 0.3–0.5 TU/cell. Cells were then washed once with PBS and processed for fluorescence microscopy as described below. At least three independent experiments, each performed in duplicate, were performed for every condition.

### 2.5. Fluorescence Microscopy

Cells washed once with PBS were fixed with 3.7% paraformaldehyde on ice for 15 min, permeabilised with 1% Triton X-100 for 10 min and stained with 4′,6-diamidin-2-phenylindol (DAPI). Fluorescence images of enhanced green fluorescent protein (EGFP)-positive cells were acquired using a BZ-9000 fluorescence microscope (Keyence Corporation, Osaka, Japan) with a 10X objective. DAPI staining was used to visualise the cell nuclei.

### 2.6. Lactose Pre-treatment for H-1PV Transduction Analysis

β-lactose was purchased from Sigma-Aldrich Chemie GmbH Darmstadt, Germany (Cat. No. L-3750). Lactose stock solution was freshly prepared before treatment of the cells. HeLa cells were seeded at a density of 4 × 10^4^ cells/well in 24-well plates and then pre-treated with increasing amounts (50, 100, 150, and 200 mM) of lactose for 30 min and then cells were infected with recH-1PV-EGFP for 4 h and grown for an additional 20 h. Cells were then processed as described in viral transduction assay and fluorescent microscopy sections. Numbers represent the arithmetic mean percentage of EGFP-positive cells relative to the number of EGFP-positive cells observed in untreated cells, which was arbitrarily set as 100%.

### 2.7. Western Blotting

Standard Western blotting was performed as described previously [15]. Immunoblotting was carried out with the following antibodies: rabbit polyclonal anti-galectin-1 (HPA000646) at 1:1000 dilution, and mouse anti-β-tubulin (T8328) (both purchased from Sigma-Aldrich, Hamburg, Germany) at 1:4000 dilution; rabbit polyclonal anti-laminin gamma 1 (PA5-36300; Thermo Fisher Scientific, Carlsbad, CA, USA) at dilution 1:1000; rabbit anti-NS1 SP8 antiserum [27] and rabbit anti-VP1/2 antiserum [28] at 1:5000 dilution. Thereafter, the membrane was incubated with horseradish peroxidase-conjugated secondary antibodies (Santa Cruz, Heidelberg, Germany) used at 1:1000 dilution.

### 2.8. Generation of the LGALS1 Knockout Cell Line

CRISPR/Cas9-mediated knockout of *LGALS1* in NCH125 was accomplished using galectin-1 Double Nickase Plasmid ([h]sc-400941-NIC), whereas the CRISPR/Cas9 negative control was obtained using the Control CRISPR/Cas9 Plasmid (sc-418922; both from Santa Cruz). NCH125 cells were seeded in a 6-well plate at about 70% confluency. After 24 h, 2 µg of DNA were transfected using Lipofectamine LTX (Thermo Fisher Scientific, Carlsbad, CA, USA) according to the vendor’s protocol. Transfected cells were selected in normal growth medium containing 1 µg/mL puromycin (Thermo Fisher Scientific, Shanghai, China) for 72 h. Individual clones were obtained by limiting dilution. Knockout was confirmed by Western blotting.

### 2.9. Plasmid Transfection

To rescue *LGALS1* expression, the plasmid encoding *LGALS1* gene was used (SC118705; OriGene Technologies, Inc. Rockville, MD, USA). NCH125 Control and LGALS1 KO cells were seeded at a density of 3x10^5^ cells/well in a 6-well plate. The next day, cells were transfected with 2.5 µg of DNA using Lipofectamine LTX or mock-transfected for 48 h.

### 2.10. Confocal Microscopy

Cells were seeded at a density of 3.5 × 10^3^ cells/spot on spot slides and grown in 50 µL of complete cellular medium. The next day, cells were infected with wild-type H-1PV at an MOI of 500 pfu/cell in a total of 70 µL of 5% fetal bovine serum (FCS)-containing medium. At 2 h post-infection, cells were fixed with 3.7% paraformaldehyde on ice for 15 min and permeabilised with 1% Triton X-100 for 10 min. Immunostaining was carried out with the following antibodies, all used at 1:500 dilution for 1 h: mouse monoclonal anti-H-1PV capsid [29] and rabbit polyclonal anti-galectin-1 (HPA000646; Sigma-Aldrich, Darmstadt, Germany). Anti-mouse Alexa Fluor 594 IgG (A11005; Thermo Fisher Scientific, Carlsbad, CA, USA) or anti-rabbit Alexa Fluor 488 IgG (A11008; Thermo Fisher Scientific, Carlsbad, CA, USA) were used as secondary antibodies. Nuclei were stained with DAPI. Images of randomly assigned cells in the green channel (galectin-1), red channel (H-1PV), or blue channel (DAPI) were acquired with a confocal microscope (Leica TCS SP5 II, Wetzlar, Germany). Picture analysis was carried out using the LAS X Software (Leica, Wetzlar, Germany).

### 2.11. MTT Viability Assay

To determine cell viability after virus infection, the conversion of 3-(4,5-dimethylthiazol-2-yl)-2,5-diphenyl-2H-tetrazolium bromide (MTT) (Sigma-Aldrich Chemie GmbH, Steinheim, Germany) was measured. For this purpose, cells were seeded on a 96-well plate at a density of 2000 cells/well in 50 μL of culture medium supplemented with 10% FCS. The next day, 50 μL of serum-free medium containing wild-type H-1PV were added on top of the cells. In rescue experiments, cells were treated with H-1PV at an MOI of 5 pfu/cell, or 5 μg/mL of recombinant galectin-1 (ab50237; Abcam, Cambridge, UK), or both simultaneously. Every 24 h post-treatment, for a total of 4 time points, 10 μL of 5 mg/mL MTT were added and subsequently incubated for 2 h at 37 °C. Thereafter, the supernatant was aspirated, and the plates were air-dried at 37 °C overnight. To solubilise the formazan product, cells were then incubated with 100 μL of isopropanol for 20 min with moderate shaking and the absorbance was read with an ELISA reader at 570 nm. Viability of treated cells was expressed as a ratio of the measured absorbance (arithmetic mean of three replicates per condition) to the corresponding absorbance of untreated cells (arbitrarily defined as 100%).

### 2.12. Binding-Only and Binding and Entry Assays

First, the culture medium was removed and replaced with 200 µL serum-free medium containing H-1PV at an MOI of 5 pfu/cell (or H-1PV at an MOI of 5 pfu/cell and 5 μg/mL of recombinant galectin-1 simultaneously in rescue experiments). Infection was performed for 1 h at 4 °C to allow only cell surface virus binding or for 4 h at 37 °C to also allow virus cell internalisation. Thereafter, cells were extensively washed with PBS, trypsinised for 5 min, quenched with serum-containing medium and subjected to three snap freeze-thaw cycles to release cell-associated viral particles. Viral DNA was purified from cell lysates using the QiAamp MinElute Virus Spin kit (Qiagen, Hilden, Germany) according to the manufacturer’s instructions. Cell-associated H-1PV genomes were quantified by following a parvovirus-specific quantitative PCR (qPCR) protocol, as previously described [12]. A minimum of three independent experiments were performed in triplicate for every condition tested.

### 2.13. Flow Cytometry

Cells were seeded at a density of 5 × 10^5^ cells/well in a 6-well plate. The next day, cells were infected with H-1PV at an MOI of 25 pfu/cell for 1 h at 4 °C. Cells were washed with ice-cold PBS and then gently scraped off with a cell lifter on ice. Cells were then fixed with 2% formaldehyde for 15 min at 4 °C and blocked with 2.5% albumin bovine fraction V (BSA; SERVA Electrophoresis, Heidelberg, Germany)/PBS for 20 min at 4 °C. Cells were then incubated with H-1PV anti-capsid antibody (dilution 1:500) for 30 min at 4 °C, and subsequently with Alexa Fluor 488 goat α-mouse (1:500) for 30 min at 4 °C. Three washes with 2.5% BSA/PBS were performed between each staining step. Analysis was carried out using a FACS Calibur (BD Biosciences, San Jose, CA, USA).

### 2.14. Tissue Microarray

Patient material for the tissue microarray was derived from paraffin embedded GBM biopsies obtained from the Department of Pathology, Haukeland University Hospital, Bergen, Norway. The project (number 2017/2505) was approved by the Regional Ethics Committee (Bergen, Norway). Control tissues (brain, liver and tonsil) were derived from autopsy material. The microarray included 61 primary GBM and 49 recurrent GBM biopsies, plus 12 biopsies from normal tissues (four each from brain, tonsil and liver). Immunohistochemical staining was carried out as described previously [30] using galectin-1 antibody (sc-166618; Santa Cruz, Santa Cruz, CA, USA) at a dilution of 1:200 followed by a biotinylated anti-mouse antibody (Vector Laboratories, Burlingame, CA, USA) at a dilution of 1:1100. Galectin-1-positive cells were counted via automated counting, as described previously [31].

### 2.15. Correlation Analysis between Gene Expression of Cancer Cell Lines and H-1PV-Induced Oncolysis

Gene expression data were taken from two databases: 53 cancer cell lines of NCI-60 dataset (https://discover.nci.nih.gov/cellminer, accessed on 23 January 2021, RNAseq data) and the Cancer Cell Line Encyclopedia (CCLE, 52 cancer cell lines) [32]. Simple linear regression models were built for 2 considered genes of interest (*LGALS1* and *LAMC1*) and 2 control genes (*LGALS3* and *GAPDH*), predicting experimentally observed EC50 values. Additionally, a two-variable regression was built predicting EC50 with both genes *LGALS3* and *GAPDH*. Significance of the models was characterised by the *p*-values (*p*), coefficients of determination (*R*^2^) and Pearson correlations (*R*).

### 2.16. Measurement of Transcript Levels

The mRNA expression levels of the target of interest *LGALS1*, as well as of the reference genes *ACTB*, *GAPDH* and *PGK1*, were quantified at the nCounter Core Facility on a SPRINT Profiler system by nCounter technology (NanoString Technologies (Seattle, WA, USA), as described previously [14]. Accession numbers and target sequences of analysed genes are:

*LGALS1* gene (accession number NM_002305.4): GGTGCGCCTGCCCGGGAACATCCTCCTGGACTCAATCATGGCTTGTGGTCTGGTCGCCAGCAACCTGAATCTCAAACCTGGAGAGTGCCTTCGAGTGCGA

*ACTB* gene (accession number NM_001101.2): TGCAGAAGGAGATCACTGCCCTGGCACCCAGCACAATGAAGATCAAGATCATTGCTCCTCCTGAGCGCAAGTACTCCGTGTGGATCGGCGGCTCCATCCT

*GAPDH* gene (NM_001256799.1): GAACGGGAAGCTTGTCATCAATGGAAATCCCATCACCATCTTCCAGGAGCGAGATCCCTCCAAAATCAAGTGGGGCGATGCTGGCGCTGAGTACGTCGTG

*PGK1* gene (NM_000291.2): GCAAGAAGTATGCTGAGGCTGTCACTCGGGCTAAGCAGATTGTGTGGAATGGTCCTGTGGGGGTATTTGAATGGGAAGCTTTTGCCCGGGGAACCAAAGC

### 2.17. xCELLigence

Cell proliferation was monitored in real time through the xCelligence system (ACEA Biosciences Inc. San Diego, CA, USA) according to the manufacturer’s instructions. Briefly, 8 × 10^4^ cells per well were seeded in a 96-well E-plate (Roche, Mannheim, Germany) in a total volume of 100 µL of complete DMEM medium. Cells were treated during the cellular growth phase with H-1PV at an MOI of 5 pfu/cell or 5 μg/mL of recombinant galectin-1, or both simultaneously. Cell proliferation was monitored every 30 min in real time. Data are expressed as “Cell index” (*n* = 3) calculated by the RTCA software 1.2.1 (Agilent) as a measure of cell adhesion and, therefore, cell viability.

### 2.18. Statistical Analysis

Results are shown as the arithmetic mean of biological replicates ± standard deviation (SD) from a representative experiment. Statistical significance was determined by a paired two-tailed Student’s *t*-test, unless stated otherwise, using Microsoft Excel 365 and/or GraphPad Prism version 8. Only values below 0.05 were considered significant: *p* ≤ 0.05 (*), *p* ≤ 0.01 (**) and *p* ≤ 0.001 (***).

## 3. Results

### 3.1. Knockdown of LGALS1, but Not LGALS3, Hampers H-1PV Infection

To identify cellular modulators of the H-1PV life cycle, we previously carried out a high-throughput siRNA library screening in cervical carcinoma-derived HeLa cells using a siRNA library targeting the human druggable genome (6961 genes, each targeted by a pool of four siRNAs/gene) [14]. This led to the identification of laminins, in particular those containing the laminin γ1 chain, as factors used by H-1PV to attach at the cell surface and to enter cancer cells [14]. In the same screening, *LGALS1*—the gene encoding Gal-1—emerged as another top activator of H-1PV transduction, as its silencing decreased H-1PV transduction by approximately 70% [14]. These findings, along with the fact that galectins interact with laminins [17], prompted us to hypothesise that Gal-1 is involved in H-1PV infection at the level of cell entry. This hypothesis was also supported by the discovery that MVM (a closely related protoparvovirus) requires Gal-3 to efficiently infect mouse cells [23,24].

We confirmed the results of the siRNA library screening by performing a knockdown of *LGALS1* using an independent siRNA. Given the role of Gal-3 in host cell entry by MVM, we also investigated the effect of siRNA-mediated silencing of *LGALS3*. A recombinant H-1PV expressing the *EGFP* reporter gene (recH-1PV-EGFP) was used for these experiments [25]. This non-replicative parvovirus shares the same capsid of the wild type but harbours the *EGFP* gene under the control of the natural P38 late promoter, whose activity is regulated by the NS1 viral protein. Therefore, the EGFP signal directly correlates with the ability of the virus to reach the nucleus and initiate its own gene transcription.

Cervical cancer-derived HeLa, glioma-derived NCH125 and pancreatic carcinoma-derived BxPC3 cell lines were transfected with siRNAs targeting *LGALS1* or *LGALS3*, or a scrambled siRNA. After 48 h, cells were infected with recH-1PV-EGFP and grown for a further 24 h. Efficient gene silencing was achieved for both genes in cells transfected with their respective siRNAs (Appendix A). However, only the siRNA targeting *LGALS1* significantly decreased H-1PV transduction (by more than 55%). As opposed to the role of Gal-3 in MVM infection, silencing of *LGALS3* did not significantly alter H-1PV transduction when compared with scramble controls (Figure 1). These results confirm the results of our siRNA library screening, which indicated that Gal-1, but not Gal-3, plays a key role in H-1PV infection in the cell lines tested.

### 3.2. Pre-Treatment with Lactose Inhibits H-1PV Infection

Gal-1 is a member of the galectins which belongs to a sub-family of lectins, defined by their highly conserved carbohydrate recognition domain (CRD) with the ability to bind to a number of beta-galactosides. Among the beta-galactosides, lactose is known to modulate the function of Gal-1 by directly binding to the CRD [33]. Hence, we hypothesised that treatment with lactose may interfere with H-1PV infectivity by competing with the virus for the interaction with Gal-1. To test this hypothesis, HeLa cells were pre-treated with increasing concentrations of soluble β-lactose before being infected with recH-1PV-EGFP. Pre-treatment with lactose decreased H-1PV transduction in a dose dependent manner (Figure 2). These results provide further evidence that H-1PV interaction with Gal-1 plays a crucial role in modulating H-1PV infectivity.

### 3.3. Galectin-1 Knockout Impairs H-1PV Infection in NCH125 Cells

To further investigate the biological role of Gal-1 in H-1PV infection, we took advantage of the CRISPR-Cas9 technology and established the NCH125 *LGALS1* knockout cell line (LGALS1 KO). We also established the NCH125 control cell line (Control) using a non-targeting guide RNA control sequence. Given that H-1PV requires S-phase factors expressed in proliferating cells for a productive infection [8], we evaluated the proliferation of LGALS1 KO versus Control cells. Both cell lines proliferated at a similar rate, as shown by real-time monitoring of cell growth and viability via xCELLigence (Appendix A).

We infected LGALS1 KO and Control cell lines with wild-type H-1PV and analysed the number of internalised virus particles by immunofluorescence using a specific anti-capsid antibody [29]. Two hours post-infection, fluorescence was significantly lower in LGALS1 KO cells than in Control cells (Figure 3A). Then, we evaluated the levels of viral proteins 48 h post-infection with the wild-type H-1PV. In agreement with previous results, Western blotting analysis revealed that NS1 and VP1 protein levels were lower in LGALS1 KO cells than in Control cells (Figure 3B). Finally, we analysed H-1PV transduction efficiency by infecting both cell lines with recH-1PV-EGFP. Consistent with reduced H-1PV entry, a significant decrease in transduction activity was found in LGALS1 KO cells (47%) in comparison with Control cells (Figure 3C). Strikingly, transient transfection of LGALS1 KO cells with a plasmid encoding *LGALS1* 48 h prior to infection, by re-establishing Gal-1 protein levels to values similar to those observed in Control cells (Figure 3C: Western blotting analysis), rescued the reduction in H-1PV transduction in these cells (Figure 3C).

### 3.4. Galectin-1 Knockout Decreases H-1PV Oncolytic Activity in NCH125 Cells

As *LGALS1* knockdown/out decreased the overall amount of internalised H-1PV, we assessed whether this would result in reduced oncolytic activity. For this purpose, we assessed the susceptibility of LGALS1 KO and Control cell lines to H-1PV infection in a time course experiment in which viability of infected cells was assessed every 24 h for a total of 96 h. Whereas the viability of Control cells decreased progressively over time, the viability of LGALS1 KO cells, which were less sensitive to H-1PV infection, remained high throughout the experiment (above 75%; Figure 4A). Remarkably, the susceptibility of LGALS1 KO cells to H-1PV oncotoxicity was re-established by infecting the cells together with human recombinant Gal-1. Indeed, the viability of LGALS1 KO cells infected with H-1PV dropped from 74% to 18% in the presence of Gal-1; a control experiment showed that the protein itself was not toxic to the cells at the concentrations used (Figure 4B). Together, these results highlight the critical role of Gal-1 in H-1PV infection in NCH125 cells.

### 3.5. Galectin-1 Plays a Role in H-1PV Virus Entry Rather Than Cell Surface Attachment

The results shown above indicate that Gal-1 is involved in the early steps of H-1PV infection. However, whether Gal-1 is required for H-1PV attachment at the cell surface, internalisation, or both events, remains to be determined. To elucidate the role of Gal-1 in H-1PV infection, we first performed virus binding and entry assays. LGALS1 KO and Control cell lines were infected with wild-type H-1PV at 37 °C for different times (0.5, 1, 2 and 4 h) and then cell-associated viral DNA was quantified by PCR. In agreement with previous results, we observed less cell-associated H-1PV DNA in LGALS1 KO cells than in Control cells (Figure 5A). The addition of recombinant Gal-1 increased the number of cell-associated H-1PV genomes in LGALS1 KO cells to values that were similar to those found in H-1PV-infected Control cells (Figure 5B). These results confirm the role of Gal-1 in H-1PV binding and/or entry.

Concerning the possible involvement of Gal-1 in H-1PV cell surface attachment, LGALS1 KO and Control cell lines were inoculated with H-1PV at 4 °C for 1 h. Under these conditions, only virus attachment at the cell surface virus occurs, while cell entry is prevented [14]. After removing unbound H-1PV particles, those that remained attached to the cell surface were stained with an anti-capsid antibody and Flow cytometry analysis was performed. No significant differences in the fluorescence signal (H-1PV binding) were observed between the LGALS1 KO and Control cells (Figure 5C). The same findings were obtained by quantitative PCR analysis (Figure 5D). These results speak for an involvement of Gal-1 in H-1PV entry rather than in H-1PV binding to the cell surface.

### 3.6. Gal-1 Cooperates with Laminins in Mediating H-1PV Infection

Given that laminins containing the γ1 chain play determinant roles in H-1PV attachment at the cell surface as well as in H-1PV cell entry, we asked whether siRNA-mediated silencing of *LAMC1* in LGALS1 KO cells would further decrease H-1PV infection (Appendix A). In agreement with the results shown above, reduced H-1PV cell uptake was observed in LGALS1 KO cells. Furthermore, given the important role of laminins in H-1PV cell attachment and entry, knockdown of *LAMC1* gene expression also strongly reduced cell-associated H-1PV genomes in NCH125 cells. The reduction observed upon *LAMC1* knockdown was not significantly different between LGALS1 KO cells and Control cells, indicating that, in the absence of Gal-1, the depletion of *LAMC1* does not further reduce H-1PV cell uptake (Figure 6A). To confirm these results, we repeated the double knock-down experiment in HeLa cells (Appendix A). To this end, we silenced *LGALS1* gene expression in the previously established HeLa LAMC1 KD cell line (in which the *LAMC1* gene was knocked down via CRISPR-Cas9) [14]. In agreement with published results, infection of HeLa LAMC1 KD cells presented a 40% reduction in cell-associated H-1PV genomes in comparison with Control cells. The silencing of *LGALS1* also decreased H-1PV cell uptake in HeLa cells, yet no significant difference was observed between HeLa control and HeLa LAMC1 KD cell lines (Figure 6B). The fact that removal/reduction of both laminin γ1 and Gal-1 does not synergistically inhibit infection suggests that the two factors may act on the same H-1PV entry pathway. At the same time, the finding that, under these conditions, a fraction of viral particles is still able to penetrate cells supports the idea that H-1PV may also use alternative pathways and exploit other cellular factors to infect cells.

### 3.7. Gal-1 Is a Marker of Bad Prognosis in Various Tumour Types including GBM

Growing evidence indicates that overexpression of *LGALS1* is associated with metastasis formation, tumour recurrence and poor tumour prognosis [34]. Our analysis of brain tumour expression datasets using the GlioVis web application (http://gliovis.bioinfo.cnio.es/, accessed on 23 January 2021) revealed that *LGALS1* overexpression is associated with worse overall survival for brain tumours. Focusing particularly on GBM, we observed that these tissues have significantly higher expression of *LGALS1* in comparison to those from healthy individuals (Appendix A) and *LGALS1* levels increase with the severity of the malignancy from WHO grade II to IV (Appendix A). High *LGALS1* expression is associated with poor prognosis in glioma (Appendix A).

Next, we investigated whether Gal-1 protein levels varied between normal tissues, primary and recurrent GBM tissues. To this end, we used an in-house protein tissue microarray including a cohort of 110 GBM patient biopsies (61 primary and 49 recurrent GBM) and 12 biopsies from normal tissues (four each from brain, liver, and tonsil) and performed immunohistochemistry using an anti-galectin-1 antibody. Levels of Gal-1 were higher overall in GBM biopsies than in normal tissues. Among the GBM biopsies, we found a diversified Gal-1 expression profile with recurrent GBM tissues expressing significantly higher levels of Gal-1 than primary GBM tissues (45% of recurrent GBM tissues expressed medium or high levels of Gal-1, compared with 20% of primary GBM tissues; Figure 7).

### 3.8. LGALS1 Expression Profile of NCI-60 Cells Positively Correlates with H-1PV Oncotoxicity

Given the important role of Gal-1 in H-1PV infection, we looked for a putative correlation between *LGALS1* expression levels and H-1PV oncotoxicity. To this end, we screened 53 cancer cell lines from the NCI-60 panel for their susceptibility to H-1PV [14]. The gene expression profiles of these cell lines are fully characterised and publicly available [35]. We assessed virus-mediated oncotoxicity by monitoring cell viability in real time using xCELLigence [14]. We calculated the viral MOI responsible for killing 50% of the cell population at 72 h post-infection (EC50). Using gene expression data from NCI-60 and the Cancer Cell Line Encyclopedia (CCLE), we found that *LGALS1* mRNA expression levels anti-correlated with EC50 values, suggesting that cells expressing higher levels of *LGALS1* may be more susceptible to virus killing activity (Figure 8).

A similar anti-correlation was previously shown for *LAMC1* [14]. Therefore, we asked whether we could better predict the susceptibility of a certain cancer cell to H-1PV infectivity and cell killing by analysing the expression levels of the two genes together. As negative controls, *LGALS3* (selected for showing no apparent role in H-1PV infection) and *GAPDH* were chosen. Using both NCI-60 and the CCLE databases, we found that a slight increase in predictability of EC50 may be achieved by combining *LAMC1* and *LGALS1* in a single linear regression model. For NCI-60, *R*^2^ of the combined model was 0.313 (*p* = 8.3 × 10^−5^), while simple models gave *R*^2^ of 0.188 (*LAMC1*, *p* = 1.2 × 10^−3^, *R* = −0.433) and 0.260 (*LGALS1*, *p* = 9.8 × 10^−5^, *R* = –0.510). Control genes show no significant linear relation (*LGALS3*: *R*^2^ = 0.015, *p* = 0.38, *GAPDH*: *R*^2^ = 0.018, *p* = 0.34) (Appendix A). Similar results were obtained using the CCLE dataset (Appendix A). These results are in line with the important role that laminin γ1 and Gal-1 play in H-1PV cell surface recognition and entry.

### 3.9. LGALS1 Expression Positively Correlates with H-1PV Oncolysis in Glioma Cell Lines

Glioma cancer cell lines are generally susceptible to H-1PV oncolysis [36]. However, not all cancer cell lines respond similarly to H-1PV oncolysis: they range from highly to lowly permissive, or are even resistant. We recently described four glioma cell lines that are semi-permissive to H-1PV infection, namely U251, LN308, T98G and A172-MG, which all express low levels of *LAMC1* mRNA [14]. However, it is possible that other cell components may account for the poor susceptibility of these cell lines to virus infection. Using the NanoString technology, we found that *LGALS1* mRNA levels were lower in the four aforementioned cell lines, as well as in the control normal human astrocytes, than in two H-1PV-sensitive glioma cell lines (NCH125 and NCH37) (Figure 9A). Monitoring of cell viability in real time by xCELLigence confirmed our previous results showing that NCH125 and NCH37 cell lines are efficiently killed by H-1PV at an MOI of 5 (pfu/cell). By contrast, U251, LN308, T98G, A172-MG cell lines as well as control normal human astrocytes were not. Remarkably, susceptibility of the four semi-permissive glioma cell lines to H-1PV oncotoxicity was substantially enhanced by the addition of human recombinant Gal-1 (Figure 9B). Consistent with our previous results, the addition of exogenous Gal-1 promoted H-1PV entry in the four glioma cell lines leading to an increase in the number of cell-associated viral genomes by 1.5- to 2.8-fold (Figure 9D), while not interfering with viral binding to the cell surface (Figure 9C). Together, these results confirm that Gal-1 plays a critical role in H-1PV infection at the level of virus entry and provide evidence that Gal-1 levels can determine the outcome of H-1PV infection. These results further support the importance of Gal-1 to H-1PV oncolytic activity and pave the way for its use in predicting the success of H-1PV infection.

## 4. Discussion

Following more than five decades of preclinical research, H-1PV monotherapy has been evaluated in patients with recurrent GBM and pancreatic carcinoma in early-phase clinical trials. H-1PV treatment was demonstrated to be safe, well tolerated and associated with surrogate evidence of anticancer efficacy, including immunoconversion of the tumour microenvironment and improved progression-free survival and overall survival in comparison with historical controls [9]. These promising results have motivated research aiming at further improving H-1PV efficacy [11,37].

The virus life cycle is a multistep process that is heavily dependent on the presence and abundance of viral (co-)receptors, processing enzymes and proteins required for a productive infection. The levels and activities of these components may vary in different cancer cells, determining their susceptibility to a particular virus. We anticipate that a better understanding of the H-1PV life cycle may provide the cues to further develop H-1PV-based therapies. For instance, this knowledge may help to identify new drugs that enhance H-1PV replication in cancer cells and/or oncolytic activity by modulating H-1PV-related cellular pathways. Furthermore, the cellular factors involved in the H-1PV life cycle may also serve as biomarkers to predict whether a certain tumour is susceptible or resistant to H-1PV infection.

Recently, we found that H-1PV enters cancer cells via clathrin-mediated endocytosis, a process that involves dynamin and requires a low pH in the endocytic compartments [15]. We also reported that laminins, in particular those containing the laminin γ1 chain, act as attachment factors at the cell surface for a successful H-1PV infection [14]. In particular, we found that sialic acid moieties in the laminins provide a docking place for the virus to anchor to at the cell surface [14]. Laminin γ1 was originally identified as a modulator of the H-1PV life cycle in a siRNA screening using a druggable-genome library performed in HeLa cells. The same screening revealed Gal-1 as another top activator of H-1PV infection. Indeed, silencing of *LGALS1* impaired H-1PV virus transduction by approximately 70% in HeLa cells. These results prompted us to explore whether Gal-1 is involved in the early steps of H-1PV infection.

In the present study, we show that Gal-1 plays a central role in H-1PV infection at the level of H-1PV cell entry but not cell attachment, indicating a role that is distinct from that of laminins in the virus cell cycle. By contrast, knockdown of *LGALS3* did not impair H-1PV infection (Figure 1), further supporting the specificity of the interaction between H-1PV and Gal-1.

Until the present study, Gal-3 was the only galectin that had been implicated in a protoparvovirus infection. Indeed, knockdown of *LGALS3* rendered LA9 mouse fibroblasts less susceptible to MVM infection. This phenotype was not due to reduced binding to the cell surface; instead, Gal-3 was responsible for promoting efficient virus uptake [24]. Our results indicate that Gal-1 is essential for a productive H-1PV infection at the level of cell entry, with no evidence of its requirement in viral binding to the plasma membrane in NCH125 cell lines. Therefore, these findings suggest that the mechanisms through which Gal-1 mediates H-1PV entry are similar to those of Gal-3 in MVM infection [23]. The fact that our results do not support an involvement of Gal-3 or clathrin-independent endocytosis in the H-1PV entry process [15] suggests that H-1PV and MVM engage different galectins for their entry processes which may contribute to their different tropism. At this point, we cannot also exclude that, in addition to Gal-1, other members of the galectin family may participate in the H-1PV entry process.

Based on our results, we envision a model in which H-1PV interacts with different classes of molecules, rather than with a single cell surface receptor, in order to enter cancer cells. Laminins containing γ1 chains would accumulate virus in the vicinity of the cell surface via sialic acid, while Gal-1 would promote the efficient internalisation of virus particles into a clathrin-coated pit. After engagement of these factors, H-1PV would penetrate the cells preferentially via clathrin-mediated endocytosis [15].

One possibility arises that H-1PV hijacks extracellular Gal-1 to enter the cells together with the protein. Our results support this idea by showing that the addition of exogenous purified Gal-1 boosts H-1PV infection at the level of virus entry, thereby sensitising semi-permissive cancer cells to H-1PV-mediated oncolysis. On the other hand, we cannot exclude that Gal-1 may play additional roles at post-entry levels, including viral trafficking and DNA uncoating.

Galectins are known to be synthesised in the cytoplasm and accumulate there until they are secreted via a poorly characterised pathway [38]. The exact mechanism through which galectin(s) translocate across the cell membrane remains also poorly understood [39]. However, previous research has shown that inhibition of the lipid raft-dependent pathway does not impede Gal-1 internalisation; instead, a total block of Gal-1 internalisation was observed only when both clathrin-mediated endocytosis and lipid rafts were disrupted, demonstrating that Gal-1 enters cells through various mechanisms, including clathrin-mediated endocytosis [40,41]. Therefore, it may be possible that H-1PV uses Gal-1 to enter cancer cells through clathrin-mediated endocytosis.

Alternatively, Gal-1 could mediate the binding of H-1PV to other cellular factors involved in its entry, e.g. a transmembrane receptor or a co-receptor. A number of studies have shown that the multivalent binding activity of Gal-1 and other galectins is able to cross-link carbohydrates and glycoconjugates [42,43]. For instance, Gal-1 cross-linking has the ability to massively redistribute a diverse population of glycoproteins on the cell surface of T cells and segregate them into membrane microdomains [44]. Gal-1 has also been associated with the assembly and remodelling of the extracellular matrix [45], and has been shown to bind to various components present there, especially those containing polylactosamine chains, such as laminins [45,46]. In this respect, *LAMC1* knockdown in NCH125 LGALS1 KO cells, or *LGALS1* knockdown in HeLa LAMC1 KD cells did not further decrease H-1PV cell uptake (Figure 6), suggesting that laminins and Gal-1 may cooperate in the early steps of H-1PV infection presumably by performing overlapping roles. However, as there is still residual internalisation of H-1PV (Figure 3C), it is likely that H-1PV may use alternative pathways to enter cells and that other unidentified cell factors are involved in this process. On the other hand, other laminins with or without the laminin-γ1 chain may contribute to residual H-1PV entry, independently from Gal-1. Future studies are needed to further characterize the pathways involved in H-1PV cell binding-entry and to determine whether other cellular factors besides laminins and Gal-1 are involved in these events.

Gal-1 and galectins in general have been described as playing important roles in different aspects of various viral infections, leading to their promotion or inhibition. For instance, Gal-1 stabilised the binding of HIV-1 to CD4^+^ T cells by cross-linking the viral gp120 and the host CD4 receptor, thereby helping HIV-1 to infect these cells [21]. Furthermore, soluble Gal-1 enhanced the uptake of HIV-1 by monocyte-derived macrophages, whereas Gal-3 had no effect on infection [47]. Enterovirus 71 is another example where Gal-1 has a supporting role. Gal-1 facilitates infection by interacting with the carbohydrate residues in VP1 and VP3 domains, leading to a more efficient release and dissemination of virus to other cells [48]. During Nipah virus (NiV) infection, Gal-1 enhances virus cell attachment to primary human endothelial cells [49]. However, later in the NiV replication cycle, Gal-1 seems to exert an inhibitory effect. Indeed, Gal-1 specifically binds to the viral glycoproteins NiV-F and NiV-G, which are responsible for cell-cell fusion and syncytia formation, thus blocking virus infection [50,51]. The inhibitory effect of Gal-1 is also observed in Influenza A infection, both in vitro and in vivo. Gal-1 binds directly to the envelope glycoproteins, stopping influenza from inducing hemagglutination and thereby impairing infectivity. Accordingly, influenza infection led to poorer survival rates in *LGALS1* KO mice than in wild-type mice [22].

Apart from their role in virus infections, galectins are also linked to apoptosis, angiogenesis, cell migration and tumour-immune escape [52]. In particular, high levels of Gal-1 are associated with cancer progression, poor prognosis and recurrence (reviewed in [34]). Several cancer types have been implicated, including gastric cancer [53], ovarian cancer [54], pancreatic cancer [55] and GBM [56]. In agreement with previous studies, our in silico analysis revealed that GBM presents significantly higher levels of *LGALS1* than normal tissues, and that *LGALS1* expression increases from grade II to IV gliomas (Appendix A). In terms of survival, high *LGALS1* expression is associated with a poor prognosis in glioma (Appendix A). To complement the bioinformatic analysis, we assessed a cohort of 122 patient biopsies by immunohistochemistry. We showed that Gal-1 protein levels vary across GBM with higher levels found in biopsies from patients with recurrent versus primary GBM, while in normal tissues the levels were relatively low (Figure 7). These findings corroborate previous studies showing that elevated levels of Gal-1 are associated with GBM progression [57,58,59]. They also further support the use of H-1PV to treat GBM, especially those cases with high Gal-1 protein content, given the key role that this protein has in virus entry and oncolysis.

We also found a correlation between *LGALS1* expression levels and the ability of H-1PV to induce oncolysis in 59 cancer cell lines (Figure 8 and Figure 9). These results suggest that tumours with elevated *LGALS1* expression levels are likely to be more susceptible to H-1PV oncolytic activity. Building on these findings, we show that, while virus attachment is unaffected, virus entry is enhanced in the U251, LN308, U87 and A172-MG semi-permissive cell lines when H-1PV is administrated together with recombinant Gal-1 (Figure 9). Under these conditions (2 h from infection), H-1PV is still likely at the very early stages of infection, strongly supporting that Gal-1 plays a role in the H-1PV cell entry process. In agreement with enhanced infection, we found that susceptibility of these semi-permissive glioma cells to H-1PV oncolytic activity increases upon addition of exogenous Gal-1, suggesting that a certain level of Gal-1 is required for an efficient and productive H-1PV infection. Together, these findings support the idea that Gal-1 may represent a limiting factor for H-1PV oncolysis, and therefore, that tumours with high Gal-1 expression are more likely to respond to H-1PV treatment.

A similar correlation was previously shown for *LAMC1* where cell lines highly expressing this gene were found to be more susceptible to H-1PV oncotoxicity [14]. Anti-correlation analysis of *LAMC1* obtained in the previous work (Pearson correlation *R* = −0.52, *R*^2^ = 0.27) stated a slightly higher anti-correlation than the one obtained in this re-analysis (*LAMC1 R* = −0.433, *R*^2^ = 0.188). Using the current datasets, *LGALS1* anti-correlation (*LGALS1*, *R* = –0.510, *R*^2^ = 0.260) is moderately higher than that of *LAMC1*. Update of the datasets with acquisition of novel mRNA sequencing data may account for these differences. Interestingly, the anti-correlation is strongest when the *LGALS1* and *LAMC1* expression levels are analysed together, suggesting that their combined expression analysis may better predict the success of H-1PV infection against a certain tumour. This is in line with the concept that both genes play an important role in H-1PV infectivity. Our results also present new scenarios for treatment in which exogenous administration of recombinant Gal-1 could constitute a promising adjunct in H-1PV-based therapies. However, given the role of Gal-1 in carcinogenesis [60], its possible use with H-1PV must be carefully evaluated.

## Figures and Tables

**Figure 1 viruses-14-01018-f001:**
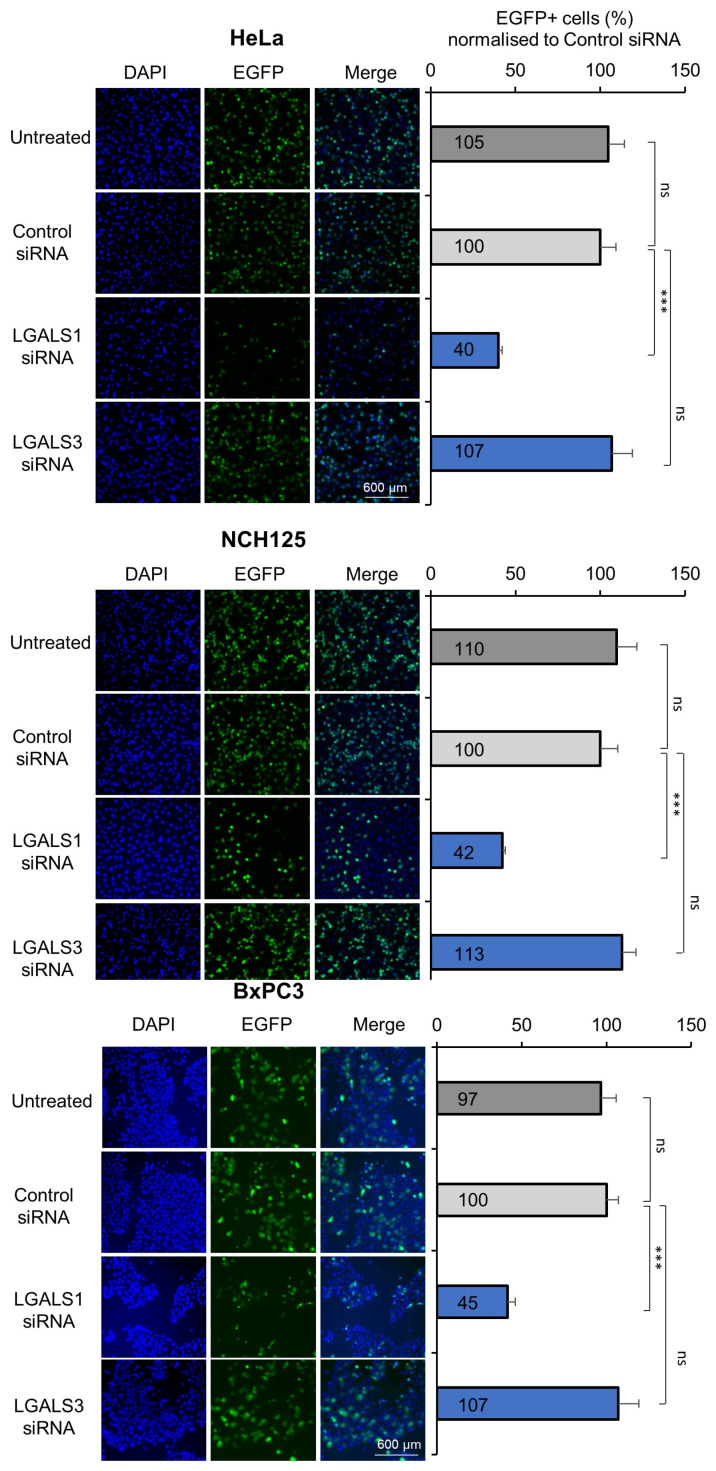
H-1PV transduction is reduced in *LGALS1*, but not *LGALS3*, knockdown cell lines. HeLa, NCH125 and BxPC3 cells were transfected with siRNAs targeting *LGALS1* or *LGALS3* or with a scrambled siRNA. At 48 h post-transfection, cells were infected with recH-1PV-EGFP for 4 h and grown for an additional 20 h. Cells were then processed as described in the Materials and Methods. Numbers represent the arithmetic mean percentage of EGFP-positive cells relative to the number of EGFP-positive cells observed in cells transfected with control siRNA, which was arbitrarily set as 100%. The independent experiment shown was repeated thrice each with three biologically independent samples (*ns–not significant*; *** *p* ≤ 0.001, calculated by using a one-way ANOVA).

**Figure 2 viruses-14-01018-f002:**
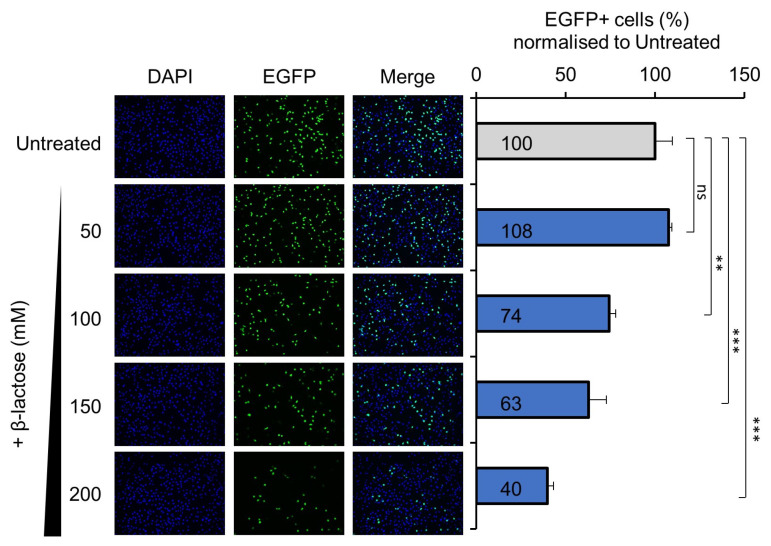
Pre-treatment with lactose decreases H-1PV infection in a dose-dependent manner. HeLa cells were pre-treated with the indicated concentrations of β-lactose for 30 min and then infected with recH-1PV-EGFP for 4 h. At 20 h after infection, cells were then harvested and processed as described in the Materials and Methods section. Numbers represent the arithmetic mean percentage of EGFP-positive cells relative to the number of EGFP-positive cells observed in untreated cells, which was arbitrarily set as 100%. The independent experiment shown was repeated thrice each with three biologically independent samples (*ns–not significant; ** p* > 0.05; *** *p* ≤ 0.001, calculated by using a one-way ANOVA).

**Figure 3 viruses-14-01018-f003:**
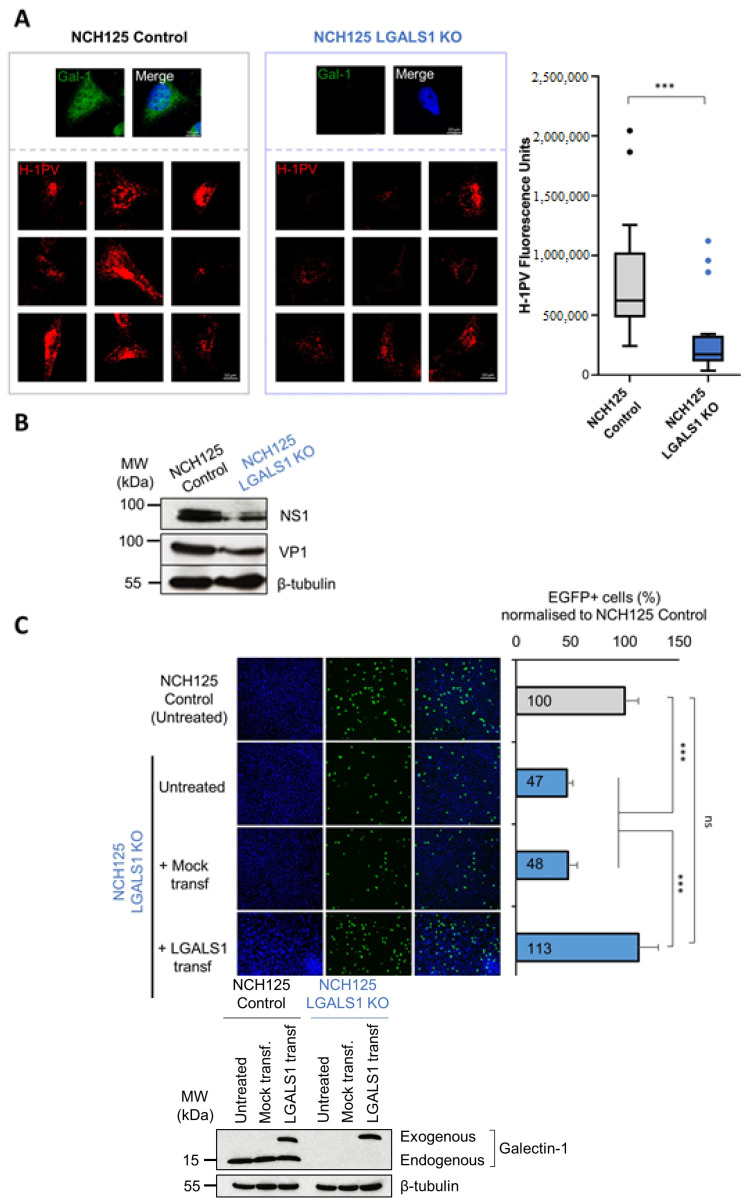
H-1PV infectivity is reduced in NCH125 LGALS1 KO cells. (**A**) H-1PV entry decreases in NCH125 LGALS1 KO cells. Control and LGALS1 KO cells were infected with H-1PV at an MOI of 50 pfu/cell for 2 h at 37 °C and prepared for confocal microscopy analysis. Gal-1 (green) and H-1PV capsid (red) were detected using specific antibodies, while DAPI (blue) was used to stain nuclei. As expected, Gal-1 was readily detected in Control cells, while it fell below detection limits in LGALS1 KO cells. The lower panel shows representative examples of H-1PV-infected cells. Quantification of the H-1PV fluorescence signal is shown on the right. This was retrieved from two independent experiments in which the fluorescence intensity was quantified in 25 randomly identified cells using ImageJ. Box plot depicts the median with a centre line, and the Tukey–Whiskers plots indicate variability outside the upper and lower quartiles (*n* = 25, *** *p* ≤ 0.001). (**B**) Control and LGALS1 KO cells were infected with H-1PV at an MOI of 2 pfu/cell for 48 h, and NS1 and VP1 protein levels were assessed by Western blotting. Beta-tubulin was used as a loading control. (**C**) H-1PV transduction is decreased in LGALS1 KO cells and re-established by transfecting the cells with a plasmid carrying *LGALS1*. LGALS1 KO cells were transfected with a plasmid encoding *LGALS1*, treated only with lipofectamine LTX (mock transfection) or left untreated. In addition, 48 h post-transfection, cells were infected with recH-1PV-EGFP for 24 h. Control cells were also included, and the level of virus transduction was set arbitrarily at 100%. The independent experiment shown was repeated twice each with four biologically independent samples (*n**s*: *p* > 0.05; *** *p* ≤ 0.001, calculating using a one-way ANOVA). On the right side, Western blotting analysis shows the levels of Gal-1 at the time of infection. Βeta-tubulin was used as a loading control.

**Figure 4 viruses-14-01018-f004:**
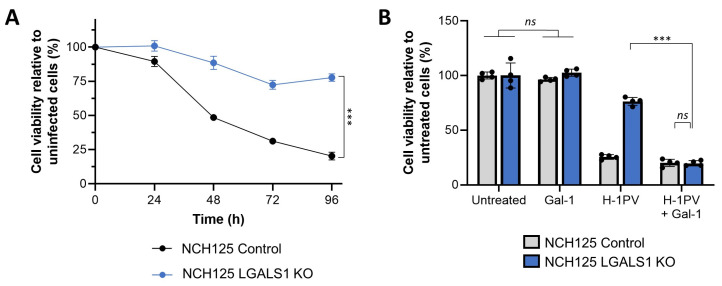
H-1PV has reduced oncolytic activity in NCH125 LGALS1 KO cells, which is rescued by supplementing with recombinant Gal-1 protein; (**A**) H-1PV oncolytic activity is reduced in NCH125 LGALS1 KO cells. Control and LGASL1 KO cells were infected with H-1PV at an MOI of 1 pfu/cell. Cell viability was assessed by MTT every 24 h for a total of 96 h. The curve plot depicts the mean ± standard deviation for each time point expressed as a percentage of cell viability compared to corresponding uninfected cells. The independent experiment shown was repeated thrice each with four biologically independent samples (*** *p* ≤ 0.001). (**B**) Purified Gal-1 rescues H-1PV oncolytic activity in NCH125 LGALS1 KO cells. Control and LGASL1 KO cells were infected (or not) with H-1PV at an MOI of 1 pfu/cell, in the presence or absence of 5 μg/mL of human recombinant Gal-1. Cell viability was assessed at 72 h post-infection by MTT. Columns depict the percentage (mean value) of cell viability compared to uninfected cells ± standard deviation bars. The independent experiment shown was repeated twice each with four biologically independent samples (*ns: p* > 0.05; *** *p* ≤ 0.001, calculated using a one-way ANOVA).

**Figure 5 viruses-14-01018-f005:**
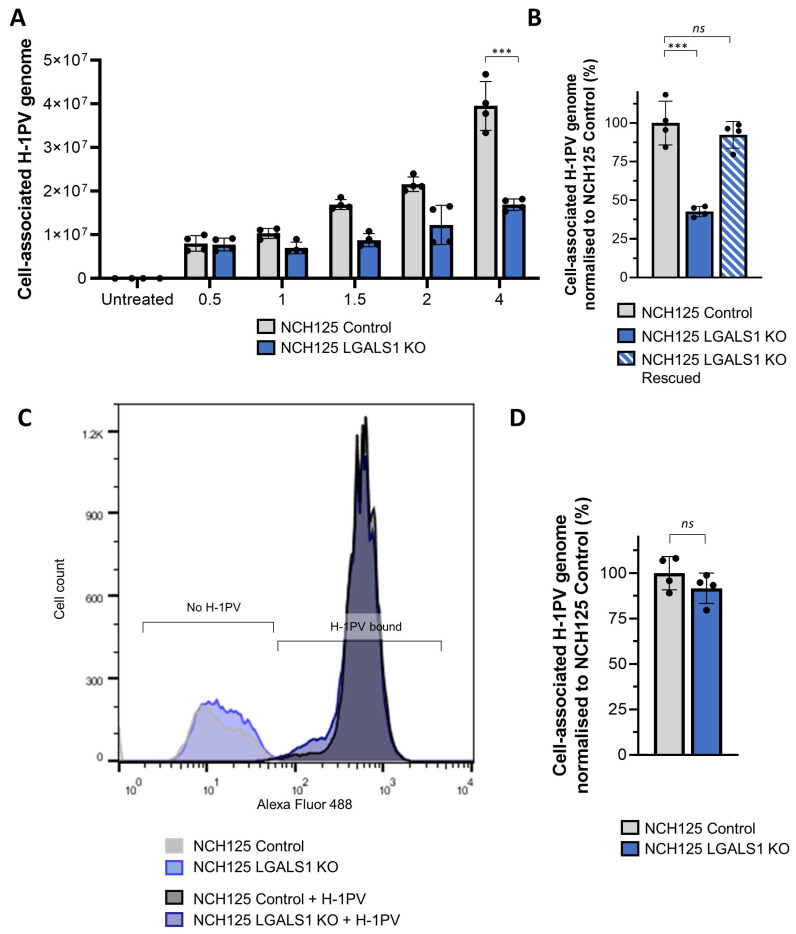
H-1PV cell entry, but not cell attachment, is reduced in NCH125 LGALS1 KO cells. (**A**) H-1PV binding and entry assay assessed by qPCR. NCH125 Control and LGALS1 KO cells were infected with H-1PV at an MOI of 5 pfu/cell for 0.5, 1, 1.5, 2 and 4 h at 37 °C. Cells were then extensively washed and harvested, and encapsidated viral DNA was extracted and subjected to qPCR. Columns in the graph show the number of copies of the cell-associated H-1PV genome with relative standard deviations (*** *p* ≤ 0.001). The independent experiment shown was repeated thrice; *n* = 3 biologically independent samples. (**B**) Binding and entry is rescued by the addition of purified recombinant Gal-1. At the time of H-1PV infection (at an MOI of 5 pfu/cell), Gal-1 was added (or not) to the culture medium. Infection was carried out for 4 h. Numbers indicate the percentage of cell-associated genomes relative to NCH125 Control cells infected with H-1PV arbitrarily set as 100% The independent experiment shown was repeated twice each with four biologically independent samples (*ns: p >* 0.05; *** *p* ≤ 0.001, calculated using a one-way ANOVA). (**C**) H-1PV cell surface binding assessed by Flow cytometry. A representative flow cytometry histogram with overlay of Control (black) and LGALS1 KO cells (blue) shows no difference in H-1PV-associated cells. Cells were either mock- or H-1PV-infected (at an MOI of 25 pfu/cell) for 1 h at 4 °C. Cells were not permeabilised for the Flow cytometry analysis, and cell surface-bound H-1PV particles were detected with a specific anti-capsid antibody. The independent experiment shown was repeated twice each with two biologically independent samples. (**D**) H-1PV binding only, assessed by qPCR. Control and LGALS1 KO cells were infected with H-1PV (at an MOI of 5 pfu/cell) for 1 h at 4 °C. Cells were then washed and harvested, and extracted encapsidated viral DNA was quantified by qPCR. The independent experiment shown was repeated thrice each with three biologically independent samples.

**Figure 6 viruses-14-01018-f006:**
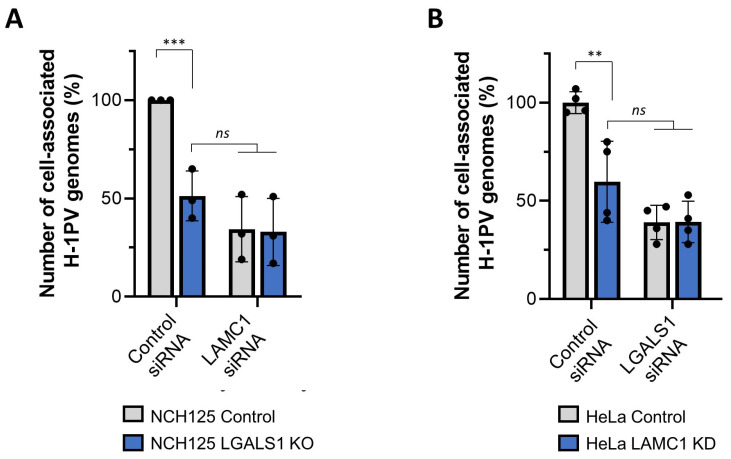
Effect on H-1PV binding and entry upon depletion of both *LAMC1* and *LGALS1* in NCH125 and HeLa cells. (**A**) NCH125 Control and LGALS1 KO cells were transfected with a siRNA targeting *LAMC1* or a negative control siRNA. At 48 h post-transfection, cells were infected with H-1PV at an MOI of 5 pfu/cell for 4 h at 37 °C. Cells were then extensively washed and harvested, and encapsidated viral DNA was extracted and subjected to qPCR. Columns in the graph show the number of copies of the cell-associated H-1PV genome with relative standard deviations (*ns*: *p* > 0.05; *** *p* ≤ 0.001, calculated using a one-way ANOVA). The independent experiment shown was repeated thrice each with threebiologically independent samples. (**B**) HeLa Control and HeLa LAMC1 KD cells were transfected with a siRNA targeting *LGALS1* or a negative control siRNA. After 48 h siRNA transfection, cells were infected with H-1PV at an MOI of 5 pfu/cell for 4 h at 37 °C. Cells were then processed as per **A**. The experiment was performed with four biologically independent samples (** *p* ≤ 0.01, calculated using a one-way ANOVA).

**Figure 7 viruses-14-01018-f007:**
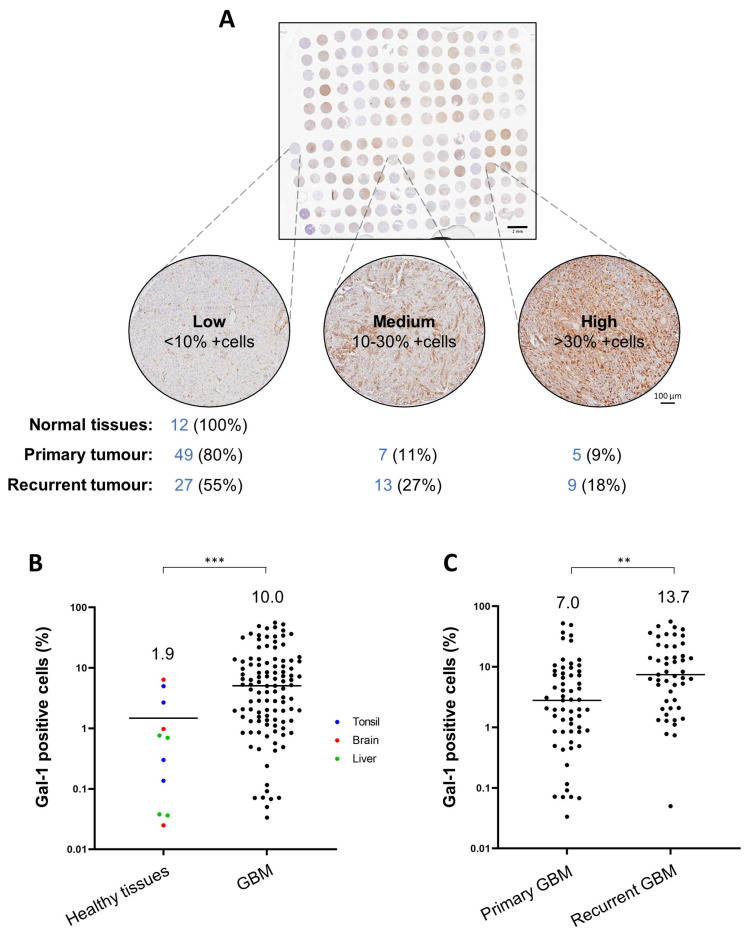
Differential expression of Gal-1 in normal tissues and in primary and recurrent GBM biopsies. (**A**) overview of the tissue microarray. This study included biopsies from normal tissue (*n* = 12), primary GBM biopsies (*n* = 61) and recurrent GBM biopsies (*n* = 49). Biopsies were categorised based on Gal-1 expression after immunostaining with anti-galectin-1 antibody: low (<10% positive cells), medium (10–30% positive cells) or high expression (>30% positive cells). The number of biopsies in each category are indicated under each representative image. The staining was performed twice in each normal sample and thrice on tumour tissues. Quantification of Gal-1-positive cells (%) was performed as described in the Materials & Methods section using in-house software; (**B**) comparative analysis of Gal-1 expression between healthy tissues and GBM biopsies (primary and recurrent); (**C**) comparative analysis of Gal-1 expression between primary and recurrent GBM biopsies. The arithmetic mean of Gal-1-positive cells is indicated with a horizontal line and by the number above (** *p*≤ 0.01; *** *p* ≤ 0.001).

**Figure 8 viruses-14-01018-f008:**
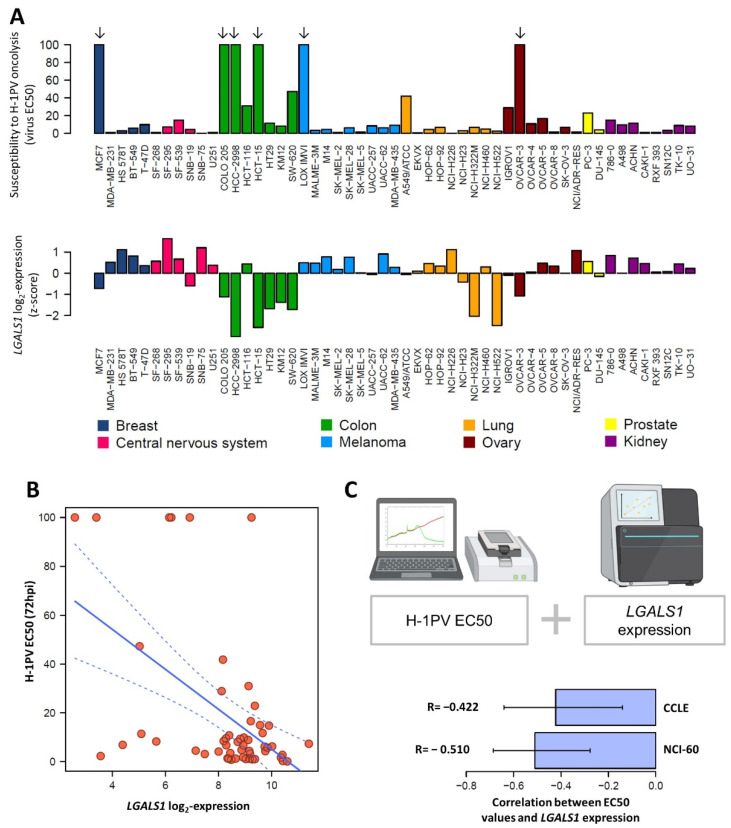
Correlation between *LGALS1* gene expression of cancer cell lines and their susceptibility to H-1PV-induced oncolysis. (**A**) *LGALS1* gene expression was retrieved from the National Cancer Institute (NCI)-60 database. Fifty-three cancer cell lines from the NCI-60 panel were tested for their susceptibility to H-1PV infection by xCELLigence. H-1PV EC50 values were calculated as the viral MOI that kills 50% of the cell population at 72 h post-infection (72hpi), measured by xCELLigence (see also [14]). Six cancer cell lines (MCF7, COLO 205, HCC-2998, HCT-15, LOX IMVI, OVCAR-3 (indicated by arrows) were found to be resistant to cell lysis even at the maximum tested concentrations of H-1PV (MOI 50 pfu/cell). Therefore, as EC50 values could not be calculated for those cell lines, their values were arbitrarily fixed as 100; (**B**) *LGALS1* expression versus H-1PV EC50. Each blue dot corresponds to a cell line and the grey line corresponds to a linear regression. (**C**) *LGALS1* levels are moderately anti-correlated with H-1PV EC50. *LGALS1* gene expression measurements were retrieved from the NCI-60 (53 cell lines) and Cancer Cell Line Encyclopedia (CCLE) (52 cell lines). Bar plot depicts the correlation between the gene expression from each dataset and the EC50 values (Pearson’s correlation). Significant anti-correlation was observed for both NCI-60 and CCLE datasets with *R* = –0.510, C.I. (–0.685, –0.277), *p* < 0.001 and –0.422, C.I. (–0.641, –0.139), *p* < 0.01 respectively (in both cases, the null hypothesis: *R* = 0).

**Figure 9 viruses-14-01018-f009:**
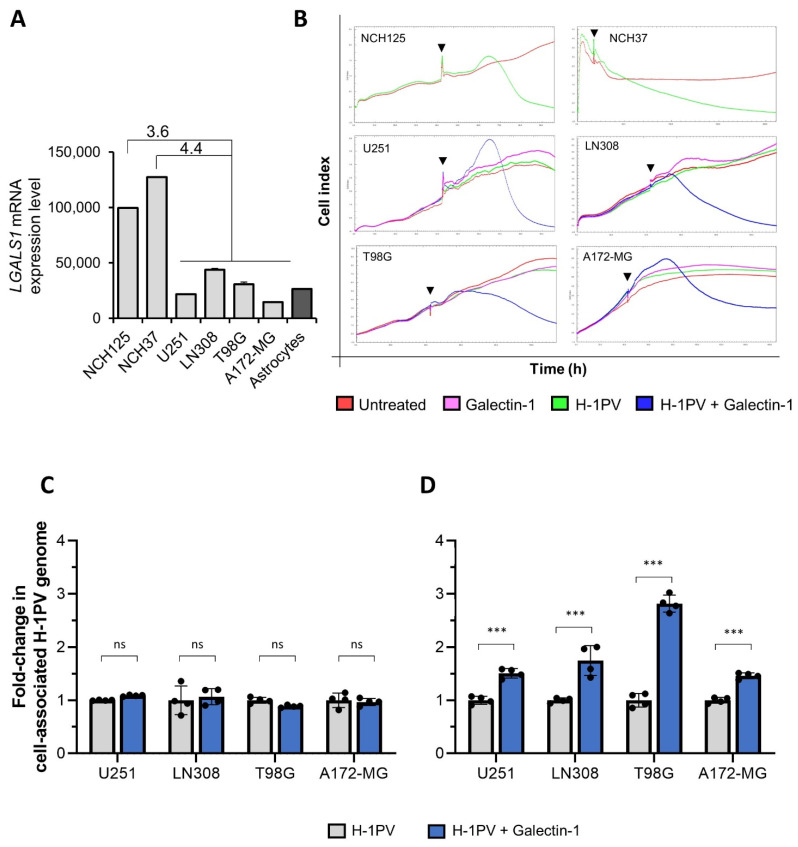
Galectin-1 levels in glioma cell lines determine the success of H-1PV infection. (**A**) Total mRNA was isolated from glioma cell lines susceptible (NCH125; NCH37) or semi-permissive (U251; LN308; T98G; A172-MG) to H-1PV infection, and *LGALS1* mRNA transcripts were measured using nCounter analysis. Bar graph depicts the *LGALS1* transcript counts; numbers on the top of the columns indicate gene expression fold changes between susceptible and semi-permissive cancer cell lines. The independent experiment is shown; *n* = 1 (NCH125, NCH37, U251 and A172-MG); *n* = 2 (LN308 and T98G); *n* = 3 (human astrocytes) biologically independent samples. (**B**) NCH125 and NCH37 cell lines were either infected with H-1PV at an MOI of 5 pfu/cell (green) or left untreated (red). Semi-permissive cell lines were infected with H-1PV at an MOI of 5 pfu/cell (green), incubated with 5 μg/mL of human recombinant Gal-1 (pink), H-1PV and Gal-1 simultaneously (blue), or left untreated (red). Cell viability was assessed by xCELLigence every 30 min in real time. The curve shows the “Cell index” mean of three biologically independent samples (*n* = 3) at any given time, which is proportional to the viability of the cell population. Black arrows indicate the time of treatment. (**C**) H-1PV binding at the cell surface is not affected by Gal-1 addition. U251, LN308, T98G and A172-MG cells were incubated with H-1PV alone at an MOI of 5 pfu/cell or with H-1PV and Gal-1. Incubations were carried out for 1 h at 4 °C (binding only). Cells were then washed and harvested, and encapsidated viral DNA was extracted and subsequently quantified by qPCR. Columns in the graph show the fold change of number of copies of cell-associated H-1PV genome relative to the virus-infected cells arbitrarily set as 1, with respective standard deviations. The independent experiment shown was performed with four biologically independent samples (*ns*: *p* > 0.05; *** *p* ≤ 0.001, calculated using a one-way ANOVA). (**D**) H-1PV entry is rescued upon Gal-1 addition. U251, LN308, T98G and A172-MG cells were incubated with H-1PV alone at an MOI of 5 pfu/cell or with H-1PV and Gal-1. Incubations were carried out either for 4 h at 37 °C (binding and entry). Cells were processed and results analysed as described in C.

## Data Availability

All relevant data supporting the findings of this study are available within the paper and its Appendix A. All other data are available from the corresponding author on request. Figure 8, Appendix A were generated employing the data sets publicly available at the CellMiner™ (https://discover.nci.nih.gov/cellminer, accessed on 23 January 2021) and at The Cancer Cell Line Encyclopedia portals (https://portals.broadinstitute.org/ccle, accessed on 23 January 2021). Appendix A were generated using the Glio-Vis data portal (http://gliovis.bioinfo.cnio.es/, accessed on 23 January 2021).

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
