# Peer review of "Oncolytic H-1 Parvovirus Hijacks Galectin-1 to Enter Cancer Cells"

_viruses, 2022, doi:10.3390/v14051018_

Round 1

Reviewer 1 Report

This study revealed that H-1PV interacts at the cell surface with galectin-1 and uses this glycoprotein to enter cancer cells. Gal-1 plays key role in the oncolytic effect of H-1PV infection in cancer cells.

Here I present several suggestions which may improve the manuscript.

  1. All the supplementary figures were not visible. It showed that the document was damaged.

  1. Fig. 1, the DAPI staining (I suppose blue represents DAPI staining) should be presented in figure 1 and the other figures. In figure legend 1, it announced that the number of EGFP+ cells observed in untreated cells was arbitrarily set as 100%, however, the figure showed that the number of EGFP+ cells in control siRNA group was 100%.

  1. Fig. 3A, line 343, control and LGALS1 KO cells were infected with H-1PV at an MOI of 50 pfu/cell, while in fig. 3B, line 352, cells were infected with H-1PV at an MOI of 2 pfu/cell, fig. 4, line 376, cells were infected with H-1PV at an MOI of 1 pfu/cell, what makes the difference in MOI? And the MOI in other figures were not mentioned.

  1. Line 465-466, Focusing particularly on GBM, we observed that these tissues have significantly higher expression of LGALS1 in comparison to those from healthy individuals, what normal tissues were used in this experiment? Line 472, normal tissues (four each from brain, liver, and tonsil) how did you get normal tissues (especially brain) from healthy individuals? Why use liver and tonsil as normal controls? And the ethics should be stated in the materials and methods part.

  1. Fig.3, the experiment was repeated twice, it should be repeated thrice. This should be complied in other experiments.

  1. How does Gal-1 mediate the entry of H-1PV into cancer cells? Can it be secreted to the microenvironment? It is really important since the exogenous Gal-1 also can help H-1PV infection.

  1. How does Gal-1 cooperates with laminins?

  1. The study lacks in vivo experiments to further prove the key role of Gal-1 in H-1PV oncolytic activity.

  1. Fig. 4A lacks the statistical P value.

  1. Fig. 9D was not mentioned in the manuscript.

Author Response

[Reviewer 1]

  1. All the supplementary figures were not visible. It showed that the document was damaged.

[Authors] – Thank you for letting us know. We will work with the assistant editor in order to make sure you receive a functioning file.

  1. 1, the DAPI staining (I suppose blue represents DAPI staining) should be presented in figure 1 and the other figures. In figure legend 1, it announced that the number of EGFP+ cells observed in untreated cells was arbitrarily set as 100%, however, the figure showed that the number of EGFP+ cells in control siRNA group was 100%.

[Authors] – Thank you for the suggestion. We have incorporated this now, and every figure displays DAPI staining only, along with EGFP and Merge. As well, the typo “untreated cells” was corrected to “cells transfected with control siRNA”.

  1. 3A, line 343, control and LGALS1 KO cells were infected with H-1PV at an MOI of 50 pfu/cell, while in fig. 3B, line 352, cells were infected with H-1PV at an MOI of 2 pfu/cell, fig. 4, line 376, cells were infected with H-1PV at an MOI of 1 pfu/cell, what makes the difference in MOI? And the MOI in other figures were not mentioned.

[Authors] – Thank you for raising this question. In Figure 3A, cells were infected with a relatively high MOI (50 pfu/cell) in order to allow visualisation of viral particles by confocal microscopy as we were unable to do so with lower viral concentrations. The same applies for Figure 5C, where a clear signal in flow cytometry was only obtained using H-1PV MOIs not lower than 25pfu/cell. For the rest of the figures, much lower MOIs have been used (between 1 to 5 pfu/cell) depending on the specific read out (virus entry, viral protein expression, oncolytic activity/cell viability) and the different time-points.

Following the suggestion of the reviewer, viral MOIs have been specified for all figures (revised figure legends).  

  1. Line 465-466, Focusing particularly on GBM, we observed that these tissues have significantly higher expression of LGALS1 in comparison to those from healthy individuals, what normal tissues were used in this experiment? Line 472, normal tissues (four each from brain, liver, and tonsil) how did you get normal tissues (especially brain) from healthy individuals? Why use liver and tonsil as normal controls? And the ethics should be stated in the materials and methods part.

[Authors] – Brain, liver and tonsils are used as standard controls in tissue microarray experiments performed by our collaborators in Norway. These tissues were obtained from autopsies. We have added the requested information including the number of the project that was approved by the Regional Ethics Committee of Bergen, (Norway) in the Materials and Methods section (paragraph 2.14)

  1. 3, the experiment was repeated twice, it should be repeated thrice. This should be complied in other experiments.

[Authors] – The reviewer 1 is correct that for Figure 3 the experiments were performed two times while for most of the other Figures, experiments were performed at least three times. In Fig. 3 A, we  analysed the intensity of the signal in 25 randomly selected cells. We then repeated the experiment obtaining very similar results. In our opinion, this analysis is sufficient to provide indications of the reduced ability of the virus to infect NCH125 KO cells in comparison to NCH125 parental cell line.  In Fig. 3 C, experiments were also carried out two times each with *four* (not three – typo corrected) biologically independent samples, analysing a large number of cells, each time for each replicate. Also in this case repetition of the experiment gave very similar results confirming the statistically significant difference between the groups analysed. Therefore we believe that these experiments support the important role of LGALS1 in H-1PV infection. These results were then further consolidated using independent methods and other cancer cell lines (other Figures of the manuscript). The number of experiments and biological replicates across all figure legends have been listed more explicitly in the revised version of the manuscript.

  1. How does Gal-1 mediate the entry of H-1PV into cancer cells? Can it be secreted to the microenvironment? It is really important since the exogenous Gal-1 also can help H-1PV infection.

[Authors] – Thank you for raising these questions. How Gal-1 mediates the entry of H-1PV into cancer cells remain to be characterized. The exact mechanisms through which galectin(s) translocate across the cell membrane are poorly understood (Elola et al., 2007). It has been described that can be secreted in the microenvironment where can accumulate at the cell surface and become a component of the extracellular matrix (Bänfer and Jacob, 2020). In the discussion section, we provide few hypotheses on how Gal-1 may mediate H-1PV entry  based on the information we found in the literature. We also explicitly state that further studies are required to elucidate the exact mechanism(s).

This paragraph was updated to incorporate more details along with one additional reference --> (Elola et al., 2007).

BÄNFER, S. & JACOB, R. 2020. Galectins in Intra-and Extracellular Vesicles. Biomolecules, 10, 1232.

ELOLA, M., WOLFENSTEIN-TODEL, C., TRONCOSO, M., VASTA, G. & RABINOVICH, G. 2007. Galectins: matricellular glycan-binding proteins linking cell adhesion, migration, and survival. Cellular and Molecular Life Sciences, 64, 1679-1700.

  1. How does Gal-1 cooperates with laminins?

[Authors] – At the moment we do not know exactly how Gal-1 cooperates with laminins in modulating H-1PV infection. Based on our results, we envisage a model in which laminins through sialic acid mediate H-1PV binding at the cell surface, while Gal-1 participates in virus cell entry. It has been described that Gal-1 has the ability to interact with a number of (glyco)proteins on the cell surface and with components of the extracellular matrix including laminins. Therefore our working hypothesis is that laminins and Gal-1 through physical interaction may form a complex (probably with some other cell factors) at the cell surface which is used by H-1PV to attach and penetrate the cells.

Please refer to the discussion section, paragraph starting on line 664.

  1. The study lacks in vivo experiments to further prove the key role of Gal-1 in H-1PV oncolytic activity.

[Authors] – We do not agree that experiments in animals are needed at this stage. Indeed, in the present study our goal was to characterize the role of Gal-1 in H-1PV cell entry and not the mechanisms underlying H-1PV-mediated oncosuppression which is surely dependent on many other factors including the role of the immune-system. We feel that the manuscript focus should remain on the mechanisms of H-1PV entry.

Nevertheless, our results in 59 cancer cell lines (53 cell lines belonging to the NCI-60 cancer panel and 6 cell lines derived by patients with GBM) reveal a direct correlation between LGALS1 expression levels and virus oncolytic activity. And this is quite conceivable as virus entry is the first important (and mandatory) step which precedes replication and oncolysis.  Thus, we feel that our results that further elucidate H-1PV entry mechanisms may be also relevant for the clinical use of H-1PV. At this stage, we believe that the discovery that Gal-1 is an important modulators of H-1PV cell entry is important enough to justify publication without additional delays and sacrificing animals. 

  1. 4A lacks the statistical P value.

 [Authors] – Thank you, it is now corrected. Indeed, there is significance as indicated (***p≤0.001).

  1. 9D was not mentioned in the manuscript.

[Authors] – Thank you for pointing this out. It is now present in the corresponding results section.

Reviewer 2 Report

In the current manuscript, the authors examine the impact of GAL1 on oncolytic parvovirus infectivity. This work is based on a previous paper in which the authors conducted an siRNA mediate screen which identified multipole factors involved in parvovirus infection. That paper looked at laminins. This paper follows up on that one by examining GAL1. Overall, the work is viewed quite positively. The finding it unique and could potentially have clinical impact. Most of the data presented is strong and easy to interpret and the manuscript itself is predominantly well written and easy to read. A few minor points (mostly concerning clarifications) are noted which should be addressed prior to publication (see below) but they do not meaningfully detract from the overall manuscript.

This reviewer does not doubt the results shown in Fig 1 and 2, however, inclusion of STDEV values would be helpful to assess the experiment to experiment variation.

While not required for publication, it would be interested to examine the NCH125 control cells transfected with LGALS1 plasmid in terms of viral infection (Fig 3C). Do these cells become more infectable if they express more LGALS1?).

Figure 4A shows a bar without significance noted (presumably the data are significant given the figure). The authors should include a marker of significance and better indicate what groups are being tested (is the test just for 96 hours or for all time points…? T-tests only compare two discrete groups so it’s unclear how this test could determine the entire data set shown to be significant)

The authors claim that the results of fig 5 “confirm the role of Gal-1 in H-1PV binding and/or entry.” (Line 396 and again in the discussion). Technically, these results only show that the impact of Gal1 is before DNA replication. GAL1 could also impact uncoating for DNA replication itself. The authors should slightly alter their text to indicate this or provide direct evidence that GAL1 alters viral entry.

Because of the overlaid nature of Fig 5C the color scheme used is very hard to visualize. The authors should change the color scheme (or go to offset histograms) to make this data more interpretable.

The data shown in Fig 6 could be interpreted many different ways and does not really support the authors conclusion that GAL1 cooperates with laminins In mediating H1-PV infection (merely that removal of both does not synergistically inhibit infection). The authors should adjust their claims accordingly.

The text for figure legend 9C is incorrect (line 565). It refers to entry while the assay described measures binding but not entry.

The test and legend for figure 8 is quite confusing. The test refers to multiple R2 values for various regressions, but the legend does not indicate which one is shown. The text should be altered to better specify what is being shown.

I question whether the data shown in Fig 8 supports the authors conclusions that LGALS1 expression correlates with oncolytic activity. A R2 value of .260 is extremely low (the correlation observed for laminin in the previous paper is .520 which is MUCH stronger) and no obvious visual correlation between EC50 and LGALS1 expression is noted by this reviewer in the data shown. I think that if the authors wish to conclude that a correlation exists they need to provide stronger evidence of such than they currently do. I might suggest examining viral infection/entry with LGALS1 expression which might provide a stronger correlation given that many additional factors could be influencing lytic activity and diluting the readout the author are currently using.

In Fig 3C the authors examine total expression of LGALS1 using western blot. However, given their overall model, cell surface LGALS1 is probably more important. The authors should probably examine the levels of cell surface GAL1 using flow cytometry to supplement their whole cell lysate results.

While not required for publication it would be interesting to examine whether treatment of virions with GAL1 increased their infectivity. The results of this type of experiment would probably allow the authors to provide a better model for their results and also have significant impact on oncolytic therapy.

Author Response

 [Reviewer 2]

In the current manuscript, the authors examine the impact of GAL1 on oncolytic parvovirus infectivity. This work is based on a previous paper in which the authors conducted an siRNA mediate screen which identified multipole factors involved in parvovirus infection. That paper looked at laminins. This paper follows up on that one by examining GAL1. Overall, the work is viewed quite positively. The finding it unique and could potentially have clinical impact. Most of the data presented is strong and easy to interpret and the manuscript itself is predominantly well written and easy to read. A few minor points (mostly concerning clarifications) are noted which should be addressed prior to publication (see below) but they do not meaningfully detract from the overall manuscript.

[Authors] – Thank you for the very positive feedback.

This reviewer does not doubt the results shown in Fig 1 and 2, however, inclusion of STDEV values would be helpful to assess the experiment to experiment variation.

[Authors] – Thank you for the suggestion. We have now incorporated the standard deviation for figures 1 and 2 in a bar graph.

While not required for publication, it would be interested to examine the NCH125 control cells transfected with LGALS1 plasmid in terms of viral infection (Fig 3C). Do these cells become more infectable if they express more LGALS1?).

[Authors] – Thank you for this suggestion. This experiment was performed only in NCH125 KO cells where transfection of the LGALS1 gene rescued H-1PV infection at levels similar to those observed in the parental cell line. We would like to keep the experiment proposed for further studies.

Figure 4A shows a bar without significance noted (presumably the data are significant given the figure). The authors should include a marker of significance and better indicate what groups are being tested (is the test just for 96 hours or for all time points…? T-tests only compare two discrete groups so it’s unclear how this test could determine the entire data set shown to be significant)

[Authors] – Thank you for pointing this out. Indeed, it was an error we made during the preparation of our manuscript and that it is now corrected. In fact, there is statistical significance as indicated (***p≤0.001). The t-test was performed at 96 hours.

The authors claim that the results of fig 5 “confirm the role of Gal-1 in H-1PV binding and/or entry.” (Line 396 and again in the discussion). Technically, these results only show that the impact of Gal1 is before DNA replication. GAL1 could also impact uncoating for DNA replication itself. The authors should slightly alter their text to indicate this or provide direct evidence that GAL1 alters viral entry.

[Authors] – Thank you very much for this comment. We believe that altogether the results presented in this study support that Gal-1 plays a role in H-1PV virus entry, like it does for other microorganisms. However, as the reviewer points out, we cannot exclude the possibility of Gal-1 playing additional roles at post-entry levels including viral DNA uncoating. Further studies are required to fully elucidate whether Gal-1 may be also involved in other steps of H-1PV life cycle. Following this comment, we have revised the discussion section to include the possibility that Gal-1 may also be required in other phases of H-1PV life cycle (lines 641-643).

Because of the overlaid nature of Fig 5C the color scheme used is very hard to visualize. The authors should change the color scheme (or go to offset histograms) to make this data more interpretable.

[Authors] – We have increased the size of the figure and the legend colours. We hope this makes the figure more legible.

The data shown in Fig 6 could be interpreted many different ways and does not really support the authors conclusion that GAL1 cooperates with laminins in mediating H1-PV infection (merely that removal of both does not synergistically inhibit infection). The authors should adjust their claims accordingly.

We agree with the reviewers that the fact that the removal of both factors does not synergistically inhibit infection is a weak argument to conclude that the two factors cooperate on the same pathway. Further studies are required to clarify this point. As suggested by the reviewer, we have adjusted our claims accordingly (lines 456-460)

The text for figure legend 9C is incorrect (line 565). It refers to entry while the assay described measures binding but not entry.

[Authors] – Thank you for pointing this out – it was a typo and is now corrected. Fig 9C refers to H-1PV binding, whereas 9D refers to H-1PV entry.

The test and legend for figure 8 is quite confusing. The test refers to multiple R2 values for various regressions, but the legend does not indicate which one is shown. The text should be altered to better specify what is being shown.

[Authors] – We have revised the text of figure legend 8. We hope that it is clearer now. 

I question whether the data shown in Fig 8 supports the authors conclusions that LGALS1 expression correlates with oncolytic activity. A R2 value of .260 is extremely low (the correlation observed for laminin in the previous paper is .520 which is MUCH stronger) and no obvious visual correlation between EC50 and LGALS1 expression is noted by this reviewer in the data shown. I think that if the authors wish to conclude that a correlation exists they need to provide stronger evidence of such than they currently do. I might suggest examining viral infection/entry with LGALS1 expression which might provide a stronger correlation given that many additional factors could be influencing lytic activity and diluting the readout the author are currently using.

[Authors] – Thank you for raising this point. Indeed, there were sources of confusion: (i) we were referring to a value from an analysis performed in 2019 and (ii) did not report Pearson correlation R, but only R2, when discussing the single-variable and two-variable linear models.  We tried to clarify this issues in the revised version of our manuscript.

First, we repeated the LAMC1 and LGALS1 gene expression- EC50 anti-correlation analysis using the current NCI-60 and UCE datasets. We confirm the anti-correlation previously found for LAMC1 (R=−0.433, p<0.001, R2=0.188) and LGALS1 (R=−0.510, p<0.001, R2=0.260) but the values found were slightly different (in the previous analysis, LAMC1 expression anticorrelates with EC50 values with a R=−0.52, p<0.001, R2=0.27, while LGALS1 anticorrelation was slightly weaker (R=–0.36 and p<0.001, R2=0.130).

For the two-variable model combining LAMC1 and LGALS1 (Supplementary Fig. S6 and S7), it is not possible to report correlation R, but only the coefficient of determination R2. The observed R2 of the combined model was 0.313 (p-value = 8.3e-5), while simple models gave R2 of 0.188 (LAMC1, p-value = 1.2e-3) and 0.260 (LGALS1, p-value = 9.8e-5). It is worth noting that using the new version of NCI-60 dataset, LGALS1 had a higher correlation than LAMC1, yet lower than both combined. 

In Fig 3C the authors examine total expression of LGALS1 using western blot. However, given their overall model, cell surface LGALS1 is probably more important. The authors should probably examine the levels of cell surface GAL1 using flow cytometry to supplement their whole cell lysate results.

[Authors] – We have performed Western blotting analysis as a control for transfection efficiency. Virus infection was performed forty-eight hours post-transfection, and therefore, produced Gal-1 had presumably been moved to the different locations. It is known that galectins localizes in the nucleus, the cytoplasm, the cell surface and in the extracellular space (Elola et al., 2007). We tried to do the control proposed by the reviewer. Unfortunately, the anti-Gal-1 antibody we used for the Western Blot analyses and that we have in the laboratory did not perform well in flow cytometry experiments.

On the other hand, we show that addition of purified Gal-1 increases virus cell entry in multiple cancer cell lines (Fig. 9D) and enhances H-1PV oncotoxicity (Fig. 9). The addition of Gal-1 was also rescuing H-1PV oncolytic activity in NCH125 LGALS1 KO.  Based on these results, it is likely that at least part of the synthesised Gal-1 upon plasmid transfection is transferred on the cell surface accounting for the increased virus transduction observed in Fig. 3C. We would like to skip the experiment proposed as we feel that we provided already sufficient evidence to support the role of Gal-1 in H-1PV virus entry. The addition of this control, may require extensive optimization including the search of an appropriate antibody and proper working dilution.

ELOLA, M., WOLFENSTEIN-TODEL, C., TRONCOSO, M., VASTA, G. & RABINOVICH, G. 2007. Galectins: matricellular glycan-binding proteins linking cell adhesion, migration, and survival. Cellular and Molecular Life Sciences, 64, 1679-1700.

While not required for publication it would be interesting to examine whether treatment of virions with GAL1 increased their infectivity. The results of this type of experiment would probably allow the authors to provide a better model for their results and also have significant impact on oncolytic therapy.

[Authors] – We have addressed this point in the sub-section 3.9 of Results: “LGALS1 expression positively correlates with H-1PV oncolysis in glioma cell lines”. In brief, the addition of exogenous Gal-1 promoted H-1PV entry in the four semi-permissive glioma cell lines (U251, LN308, T98G and A172-MG) leading to an increase in the number of cell-associated viral genomes (1.5- to 2.8-folds) and enhanced oncolysis (Figure 9D).

Reviewer 3 Report

In the present manuscript, Ferreira et al. demonstrate the role of galectin-1 (Gal-1) in parvovirus (H-1PV) entry.
Galectin-1 was identified in an siRNA screen for H-1PV host factors. To validate its role in H-1PV infection, siRNA knockdown was performed, showing reduced H-1PV mediated GFP expression upon Gal-1 knockdown in three different cell lines. 
Since a role for Gal-3 has been reported previously for MVM, the authors also ruled out a contribution of this molecule to H-1PV entry.
Next, a lactose competition assay was performed, showing that lactose (which binds Gal-1) inhibits H-1PV infection in a dose-dependent manner. Gal-1 knockout cells were established to further analyze the role of Gal-1 in H-1PV infection. Viral protein expression was reduced in knockout cells and could be rescued using a plasmid for expression of Gal-1. H-1PV replication and cytotoxicity were also reduced in knockout cells. 
Experiments with H-1PV inoculation at 4°C (allowing attachment, but not entry) and 37°C showed that Gal-1 is involved in cell entry, since virus attachment was not impaired in Gal-1 knockout cells. 
The authors had previously demonstrated that LAMC1 is also crucial for H-1PV entry. Knockout/knockdown of both LAMC1 and Gal-1 indicated that both factors act in the same step of H-1PV entry, since there were no significant differences between cells with one or both factors missing.
Next the authors addressed how Gal-1 may affect cancer therapy with H-1V, since this virus has been investigated in clinical trials for treatment of glioblastoma. Interestingly, Gal-1 was overexpressed in glioblastoma compared to healthy tissue. A correlation between Gal-1 expression and sensitivity to H-1PV (as determined by EC50) was observed in the NCI60 panel. Gal-1 increased the sensitivity of several semi-permissive glioma cell lines to H-1PV oncolysis. 

Identification of viral entry factors is important for basic understanding of virology and to develop antiviral strategies. In case of H-1PV, host factors are of specific interest since this agent is currently investigated as an oncolytic virotherapy against different malignancies. 
Overall this is a well conducted study. The experiments follow a logical sequence, all necessary controls are included and the results are described clearly.

I have a few (mostly minor) questions/comments:

- Number of replicates/statistics: To me in many instances the number of replicates for each experiment and statistics were not quite clear. The methods indicate that at least three independent experiments were performed, but the representation is not clear to me. 
For instance in Figure 1, is this a single experiment or representative of how many replicates? Were the percentages derived from quantification of one well or several wells, from one experiment or several?
Along the same line, statements such as "the independent experiment shown was repeated twice, n = 2 biologically independent samples" (line 423) I found confusing.
The methods section indicates that paired two-tailed Student's t test was performed unless stated otherwise, but for comparison of several conditions as in Figure 1, t test is not appropriate and ANOVA should be used instead.
Please check and amend these points throughout the entire manuscript. 
Also, perhaps use "(arithmetic?) mean" instead of "average" (e.g., line 262). 

- In Figures 1 and 2, perhaps present the percentages as bar graphs including statistics/SD. 

- Figure 3: The methods indicates that knockout cell lines were obtained by limiting dilution, thus these are probably single cell clones. Nevertheless, there seems to be quite some variation between individual cells and H-1PV infection is not completely abolished about Gal-1 knockdown. Please comment on this (e.g., in the discussion). 

- Figure 3: The legend states that fluorescence intesity was quantified in 25 cells. Were these cells selected randomly and was the researcher blinded to experimental conditions? Perhaps include a brief statement on this in the methods.

- (Optional): Perhaps the authors would like to summarize their model of H-1PV entry as outlined in the discussion in a graphical representation?

- Some (minor) language editing may be required, some examples: 
line 135: fluorescence microscopy
line 211: actually not cell sorting, but flow cytometry
line 399: perhaps "inoculated" rather than "infected", since infection should not occur at 4°C
line 470: "in-house" instead of "home made"?
line 564: "at any given time in proportion with to the viability": please check this sentence
line 649-652: long sentence; perhaps rephrase to make it easier to read

Author Response

[Reviewer 3]

Identification of viral entry factors is important for basic understanding of virology and to develop antiviral strategies. In case of H-1PV, host factors are of specific interest since this agent is currently investigated as an oncolytic virotherapy against different malignancies.

Overall this is a well conducted study. The experiments follow a logical sequence, all necessary controls are included and the results are described clearly.

[Authors] – Thank you for the very positive feedback.

- Number of replicates/statistics: To me in many instances the number of replicates for each experiment and statistics were not quite clear. The methods indicate that at least three independent experiments were performed, but the representation is not clear to me.

For instance in Figure 1, is this a single experiment or representative of how many replicates? Were the percentages derived from quantification of one well or several wells, from one experiment or several?

Along the same line, statements such as "the independent experiment shown was repeated twice, n = 2 biologically independent samples" (line 423) I found confusing.

[Authors] – Thank you for pointing this out. We have now clearly indicated how many times experiment was repeated (and how many biological replicates were performed for each experiment) in the corresponding figure legend.

The methods section indicates that paired two-tailed Student's t test was performed unless stated otherwise, but for comparison of several conditions as in Figure 1, t test is not appropriate and ANOVA should be used instead.

[Authors] – Thank you for noting this. It is now corrected for Figure 1 as well as for the rest of the figures where more than 2 groups have beencompared.

- In Figures 1 and 2, perhaps present the percentages as bar graphs including statistics/SD.

[Authors] – Thank you for pointing this out. Figure 2 has statistics now. Figures 1 and 2 have now bar graphs alongside.

- Figure 3: The methods indicates that knockout cell lines were obtained by limiting dilution, thus these are probably single cell clones. Neverthelerss, there seems to be quite some variation between individual cells and H-1PV infection is not completely abolished about Gal-1 knockdown. Please comment on this (e.g., in the discussion).

[Authors] – Please refer to the following paragraph in the discussion: “Yet, as there is still residual internalisation of H-1PV (Figure 3C), it is likely that H-1PV may use alternative pathways to enter cells and that other unidentified cell factors are involved in this process.” (lines 674-680). To make the connection clearer, we have referenced Figure 3C.

- Figure 3: The legend states that fluorescence intesity was quantified in 25 cells. Were these cells selected randomly and was the researcher blinded to experimental conditions? Perhaps include a brief statement on this in the methods.

[Authors] – Thank you for the suggestion. The researcher performed this randomly and this detail is now stated in the respective figure legend and in the Materials and Methods section.

- (Optional): Perhaps the authors would like to summarize their model of H-1PV entry as outlined in the discussion in a graphical representation?

[Authors] – Thank you for the suggestion. We now provide a graphical abstract illustrating our model.

- Some (minor) language editing may be required, some examples:

perhaps use "(arithmetic?) mean" instead of "average" (e.g., line 262).

line 135: fluorescence microscopy

line 211: actually not cell sorting, but flow cytometry

line 399: perhaps "inoculated" rather than "infected", since infection should not occur at 4°C

line 470: "in-house" instead of "home made"?

line 564: "at any given time in proportion with to the viability": please check this sentence

line 649-652: long sentence; perhaps rephrase to make it easier to read

[Authors] – Thank you, all points have been corrected.

Round 2

Reviewer 1 Report

The authors still did not answer question 4, "Why use liver and tonsil as normal controls? " Nevertheless, a brief explanation can be added in the material and methods part before the manuscript is published. The rest questions are well answered and no revision is needed.